# Dynamically controlled Purcell enhancement of visible spontaneous emission in a gated plasmonic heterostructure

Yu-Jung Lu[1,2], Ruzan Sokhoyan [1], Wen-Hui Cheng [1], Ghazaleh Kafaie Shirmanesh[1], Artur R. Davoyan[1,3,4], Ragip A. Pala[1], Krishnan Thyagarajan[1] & Harry A. Atwater[1,3]

Emission control of colloidal quantum dots (QDs) is a cornerstone of modern high-quality lighting and display technologies. Dynamic emission control of colloidal QDs in an optoelectronic device is usually achieved by changing the optical pump intensity or injection current density. Here we propose and demonstrate a distinctly different mechanism for the temporal modulation of QD emission intensity at constant optical pumping rate. Our mechanism is based on the electrically controlled modulation of the local density of optical states (LDOS) at the position of the QDs, resulting in the modulation of the QD spontaneous emission rate, far-field emission intensity, and quantum yield. We manipulate the LDOS via field effect-induced optical permittivity modulation of an ultrathin titanium nitride (TiN) film, which is incorporated in a gated $TiN/SiO_2/Ag$ plasmonic heterostructure. The demonstrated electrical control of the colloidal QD emission provides a new approach for modulating intensity of light in displays and other optoelectronics.

[1] Thomas J. Watson Laboratories of Applied Physics, California Institute of Technology, Pasadena, CA 91125, USA. [2] Research Center for Applied Sciences, Academia Sinica, Taipei 11529, Taiwan. [3] Kavli Nanoscience Institute, California Institute of Technology, Pasadena, CA 91125, USA. [4] Resnick Sustainability Institute, California Institute of Technology, Pasadena, CA 91125, USA. Yu-Jung Lu and Ruzan Sokhoyan contributed equally to this work. Correspondence and requests for materials should be addressed to H.A.A. (email: haa@caltech.edu)

Tailoring light emission of quantum emitters, such as semiconductor quantum dots (QDs), is a central theme of nanotechnology[1]. The spontaneous emission decay rate of a solid state emitter can be modified by coupling the emitter to a nanostructured environment with a tailored local density of optical states (LDOS)[2,3]. Previously this phenomenon, also known as the Purcell effect, has yielded a 540-fold increase of the emission decay rate and a simultaneous 1900-fold increase of total emission intensity for colloidal QDs coupled to a plasmonic nanocavity[4]. Typically, the properties of the nanostructured environment are fixed at the time of fabrication, which also sets the spontaneous emission decay rate of the emitter. Conventionally, the power radiated from an array of quantum emitters is dynamically modulated (actively controlled) by changing the optical[5] or electrical[6] pump intensity within a given nanostructured environment. On the other hand, the coupling of emitters to a local environment with tunable optical properties, as done in this study, enables the modulation of the emitter decay rate while keeping the optical pump power constant. Previous experiments have demonstrated the tunability of the narrowband emission of an epitaxial QD, which is coupled to a high-quality factor dielectric cavity[7,8]. However, these experiments can be performed only at cryogenic temperatures, making them less amenable for immediate practical applications.

Recently, there has been considerable interest in studying active nanophotonic structures with tunable optical responses[9]. Active plasmonic structures are especially interesting candidates for tuning the emission of room temperature broadband solid state emitters. This is due to the fact that they offer small optical mode volumes and relatively low-quality factors, thus eliminating the necessity of careful alignment of cavity and emitter resonances. Previous works have reported field-effect tuning of the carrier density and Fermi level in graphene[10], transparent conducting oxides[11], and silicon[12] to modulate transmittance and reflectance of plasmonic structures in the near- and mid-infrared wavelength range. Previous work has adopted this tuning mechanism to electrically modulate the 1.5 μm wavelength emission of trivalent erbium ions coupled to a graphene sheet[13]. In an alternative approach, a previous study has used the optically induced phase transition in vanadium dioxide to modulate near-infrared emission of erbium ions coupled to Salisbury-screen-type

heterostructure[14]. However, field effect modulation of the optical response of a nanophotonic structure has not been demonstrated at visible wavelengths. This is primarily due to the relatively low carrier concentrations of previously used doped oxides and semiconductors[11,12] that limit their epsilon-near-zero (ENZ) wavelengths to the near- or mid-infrared wavelength range.

Here we use degenerately doped n-type titanium nitride (TiN)[15–20], with an ENZ wavelength in the visible spectrum, to demonstrate the field effect tunable optical response of a gated TiN/SiO$_2$/Ag heterostructure at visible wavelengths. We embed InP/ZnS core–shell colloidal QDs in the SiO$_2$ layer of the tunable TiN/SiO$_2$/Ag heterostructure and study emission properties of the QDs while biasing the TiN and Ag layers with respect to each other. We observe a temporal modulation of the QD spontaneous emission rate, far-field emission intensity, and quantum yield that occurs due to the modulation of the LDOS at the position of the QDs. The optoelectronic device fabricated in this work exemplifies how conventional electronic components, in our case, a metal-oxide-semiconductor capacitor, can be adapted to the field of nanophotonics to yield the bias-controlled modulation of the PL intensity of fluorophores. Our proof-of-principle experiment demonstrates an active plasmonic mechanism for modulating visible light that is extensible to other types of quantum emitters.

## Results

**Field effect modulation of dielectric permittivity of TiN.** The schematic of our TiN/SiO$_2$/Ag heterostructure is shown in Fig. 1a. As seen in the high-resolution transmission electron microscopic image, the fabricated heterostructure consists of an optically thick Ag, a 9 nm-thick insulating SiO$_2$ spacer, and a 7 nm-thick conductive TiN layer (Fig. 1b). Visible-emitting colloidal QDs with diameters of 4–5 nm are embedded in the insulating SiO$_2$ spacer (Fig. 1c). In our work, we use InP/ZnS core–shell colloidal QDs (hereafter, InP QDs), which are of greater application interest because they are free of heavy metals and may be beneficial in considering health and environmental concerns. When a bias is applied between TiN and Ag, a charge depletion or accumulation layer is formed in the TiN at the TiN/SiO$_2$ interface (Fig. 1d and Supplementary Fig. 1). This results in a modulation of the complex refractive index of TiN and, therefore, also modulation of the LDOS at the position of QDs embedded in the SiO$_2$ layer.

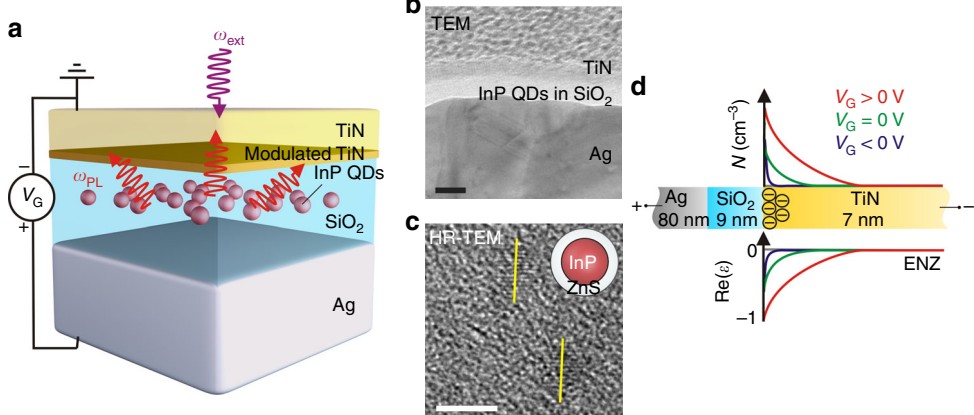

**Fig. 1** Concept of gated TiN/SiO$_2$/Ag plasmonic heterostructure for active control of spontaneous emission. **a** Schematic of the gated plasmonic heterostructure that consists of 80 nm-thick Ag and 9 nm-thick SiO$_2$ layers in which InP quantum dots (QDs) are embedded, followed by a 7 nm layer of TiN. The filling factor of the QDs in SiO$_2$ is 9%. **b** Cross-sectional transmission electron microscopy image of the fabricated heterostructure. The image shows that the deposited TiN film is conformal and smooth. The scale bar is 10 nm. **c** High-resolution transmission electron microscopic image of InP QDs with the diameter of 4–5 nm. The PL emission of the QDs peaks at 630 nm. The scale bar is 5 nm. **d** The proposed physical mechanism of modulation of optical response. When Ag is biased positively (negatively) with respect to TiN, a charge accumulation (depletion) layer is formed in TiN at the TiN/SiO$_2$ interface. Charge accumulation reduces the real part of the dielectric permittivity of the TiN

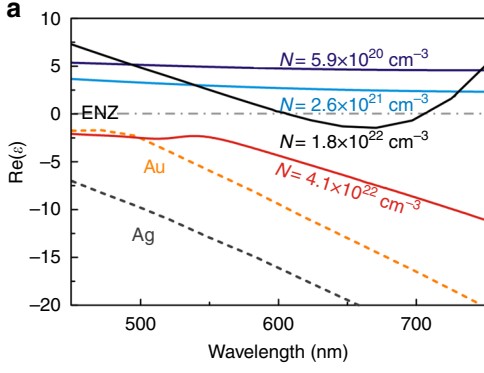

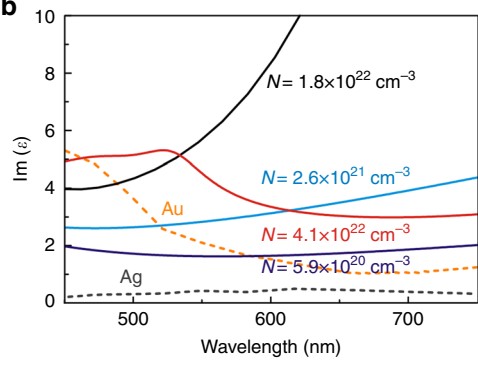

**Fig. 2** Optical properties of TiN films. Measured **a** real and **b** imaginary parts of the complex dielectric permittivity of TiN thin films. The fabricated TiN films are n-type with carrier densities ranging from $5.9 \times 10^{20}$ to $4.1 \times 10^{22}$ cm$^{-3}$. Depending on the carrier density, the fabricated TiN films can be optically dielectric (Re($\varepsilon$) > 0) or optically plasmonic (Re($\varepsilon$) < 0). The gray dotted line in **a** denotes Re($\varepsilon$) = 0. For a carrier density of $1.8 \times 10^{22}$ cm$^{-3}$, Re($\varepsilon$) is in the epsilon-near-zero (ENZ) region (–1 < Re($\varepsilon$) < 1) over the entire wavelength range. By applying a voltage between the TiN and Ag (Fig. 1), the interfacial TiN film region undergoes a transition from an optically dielectric to an optically plasmonic state or vice versa. For comparison, we also plot the dielectric permittivity values for gold and silver (Johnson and Christy)[31], two standard plasmonic materials, which are shown in dashed lines

Depending on the deposition conditions, TiN can either be in an optically dielectric phase (Re($\varepsilon$) > 0) or in an optically plasmonic phase (Re($\varepsilon$) < 0) in the visible wavelength range (Fig. 2). Moreover, one can identify a carrier concentration that yields ENZ-TiN (–1 < Re($\varepsilon$) < 1), which is crucial for observation of reflectance modulation (Supplementary Fig. 4).

**Dynamic control of emission rate of QD via LDOS modulation.** The optical measurements performed on the ENZ-TiN/ SiO$_2$/Ag heterostructures at the QD emission wavelength of $\lambda =$ 630 nm show a relative reflectance modulation of 18% (normalized to the reflectance at zero bias) when the gate voltage is varied between –1 and +1 V (Fig. 3a). Additionally, we demonstrate that the device has a modulation speed exceeding 20 MHz (Supplementary Fig. 5). The reflectance is modulated by changing the carrier density under field effect gate control in the TiN layer. This in turn modulates the real part of dielectric permittivity of the TiN from positive to negative values through the so-called ENZ regime, which marks the borderline between dielectric and plasmonic media (Fig. 3b). Moreover, when we replace the top TiN electrode by a titanium (Ti) electrode of identical thickness, we observe no reflectance modulation under an applied bias (Fig. 3c, d). This shows that the TiN plays a crucial role in the

observed optical modulation (for reflectance calculations, see Supplementary Fig. 6).

Having established gate-tunable reflectance, we study emission modulation of InP QDs embedded in the SiO$_2$ spacer layer of the TiN/SiO$_2$/Ag plasmonic heterostructure (Fig. 1a). The dependence of the photoluminescence (PL) intensity on applied bias at the QD peak emission wavelength of $\lambda =$ 630 nm is shown in Fig. 4a (for optical setup, see Methods). As seen in Fig. 4a, the PL intensity monotonically increases with positive bias and monotonically decreases with negative bias. At $\lambda =$ 630 nm, the PL intensity change (normalized to the PL intensity at zero bias) is as high as 15% when gate voltage is varied between –1 and +1 V. On the other hand, for QDs embedded in the Ti/SiO$_2$/Ag passive heterostructure (the same device was used for Fig. 3c), no PL intensity modulation is observed under applied bias (Fig. 4b). In our experiments, we observe no shift of the wavelength of PL peak intensity under applied bias in both cases when QDs are embedded in a tunable TiN/SiO$_2$/Ag or passive Ti/SiO$_2$/Ag heterostructure (Supplementary Figs. 7 and 8). Hence, at applied electric fields of 1.1 MV cm$^{-1}$, our InP QDs show no quenching or red-shift of emission, which is characteristic for cadmium-based core–shell colloidal QDs[21,22]. This is attributable to the large bandgap difference between InP core and ZnS shell materials[21]. Thus the electron and hole wave functions are well confined to the core of the core/shell QDs with an external electric field. The fact that we observe both an increase and decrease of PL intensity, depending on the polarity of the electric field, provides additional evidence that the observed modulation of PL intensity is not caused by the change of the internal state of the QDs under applied bias. If this were the case, the observed modulation of the PL intensity would depend only on the magnitude of the electric field and not on its direction.

The measured PL intensity spectrum shows significant modulation only around the central emission wavelength of $\lambda$ = 630 nm. This seems to contradict the theoretical prediction of the broadband LDOS modulation under applied bias (Fig. 4d). The apparent contradiction is resolved by recalling that the measured PL intensity spectrum originates from an inhomogeneously broadened QD ensemble. Note that the relative PL intensity modulation of each QD does not depend on the emission wavelength (Supplementary Fig. 7c). In the measured ensemble, the size distribution of the QDs is such that a large fraction of the individual QDs emits around the wavelength of $\lambda$ = 630 nm, yielding a brighter PL signal at $\lambda =$ 630 nm.

Once we have demonstrated the PL intensity modulation of the QDs, we perform time-resolved PL measurements to identify the lifetime of the QDs embedded in the TiN/SiO$_2$/Ag heterostructures. When no electrical bias is applied, the measured lifetime of the InP QDs is 390 ps. This shows that embedding the QDs into the TiN/SiO$_2$/Ag plasmonic heterostructure results in 28-fold reduction of the QD lifetime as compared to the lifetime of the QDs on bare silicon (Supplementary Fig. 10). At applied electrical bias of +1 V, the lifetime of the QDs decreases by 12%, while at the bias of –1 V, the lifetime increases by 18% (Fig. 4c). Our calculations indicate that the optical frequency electric field radiated by a QD is tightly confined in the SiO$_2$ layer and shows a considerable enhancement at the interface with TiN (Fig. 4d). As a result, the LDOS in the middle of the SiO$_2$ layer is sensitive to the modulation of the complex refractive index of the TiN film (Supplementary Fig. 11). The observed lifetime as well as the PL intensity and reflectance modulation reported in our work cannot be fully explained by changing the carrier concentration in the Drude term of the Drude–Lorenz model that describes the dielectric permittivity of TiN. At high carrier concentrations ($N$ = $1.8 \times 10^{22}$ cm$^{-3}$ for ENZ-TiN), a number of different effects may contribute to the observed optical modulation, such as the

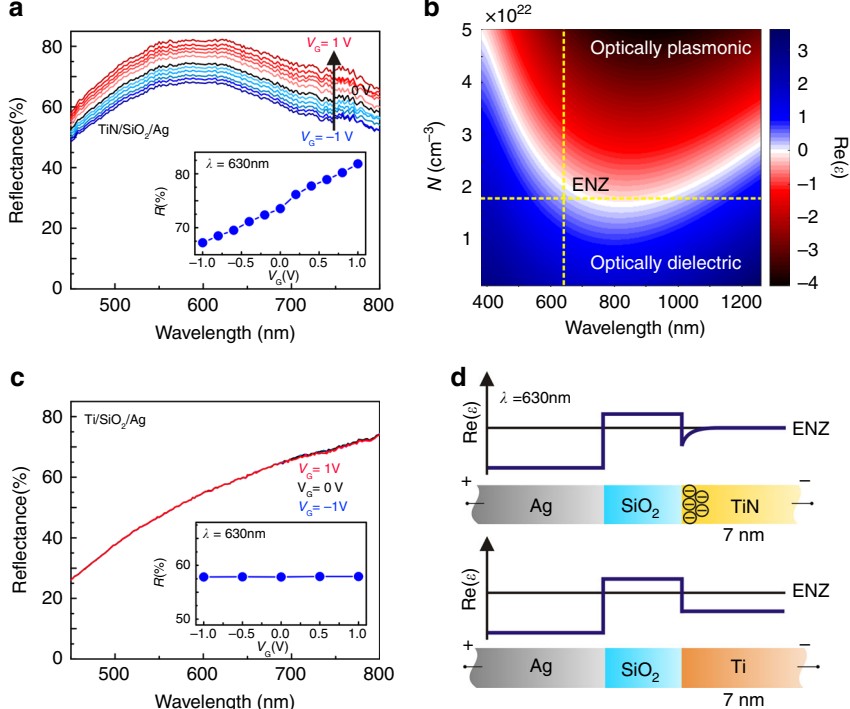

**Fig. 3** Reflectance modulation under applied gate bias: field-effect induced dielectric permittivity change in TiN. **a** Measured reflectance spectrum of a gated TiN/SiO$_2$/Ag plasmonic heterostructure for different applied voltages. Here Ag is biased with respect to TiN, and the applied voltage varies from −1 to 1 V in 0.2 V steps. The inset of **a** shows the heterostructure reflectance as a function of voltage for a wavelength of $\lambda = 630$ nm. The reflectance increases from 67 to 82% when the applied bias varies from −1 to 1 V. **b** Calculated real part of the TiN dielectric permittivity as a function of wavelength and carrier concentration. The vertical dashed line shows the operation wavelength of our device ($\lambda = 630$ nm), while the horizontal dashed line marks the carrier concentration of TiN. The two dashed lines intersect in the epsilon-near-zero (ENZ) region of the dielectric permittivity of TiN, indicating that under an applied bias TiN undergoes a transition from an optically dielectric to an optically plasmonic phase or vice versa. **c** Measured reflectance spectrum of the gated Ti/SiO$_2$/Ag control heterostructure. The inset of **c** shows the reflectance of the heterostructure as a function of electrical bias applied between Ti and Ag for a wavelength of $\lambda = 630$ nm. No visible reflectance modulation of the Ti/SiO$_2$/Ag control heterostructure is observed when the applied bias varies between −1 and 1 V. **d** Spatial distribution of Re($\varepsilon$) in the designed heterostructures. In the TiN/SiO$_2$/Ag heterostructures, the carrier density of TiN is lower than in metals, so the direct-current (DC) electric field penetrates into the TiN film, resulting in a graded spatial variation of epsilon. In the Ti/SiO$_2$/Ag control heterostructures, the DC electric field is unable to penetrate into the metallic electrodes due to the inherently higher metallic Ti carrier density. Hence, the Re($\varepsilon$) exhibits an abrupt change at the Ti/dielectric interface

dependence of the electron effective mass on the applied voltage due to the nonparabolicity of the conduction band and the dependence of the electron mobility on the applied bias. In our theoretical calculations (Fig. 4d), we assume that the dielectric permittivity of the 1 nm-thick accumulation layer of TiN is given by the measured dielectric permittivity of the TiN film with carrier concentration of $N = 4.1 \times 10^{22}$ cm$^{-3}$ (Fig. 2). Strictly speaking, this assumption is not accurate; however, it allows us to estimate the sensitivity of the calculated LDOS with respect to the variation of the refractive index of the 1 nm-thick accumulation layer (Fig. 4d).

To summarize, in our experiment we observe either a simultaneous increase of the PL intensity and total decay rate of emission $\Gamma_{tot}$ or a simultaneous decrease of the PL intensity and $\Gamma_{tot}$. However, the observed PL intensity modulation it not necessarily caused the actively changed LDOS. For example, variation of excitation field intensity under applied bias could also result in modulation of PL intensity. To investigate this possibility, we measure the absorbance spectrum of the ENZ-TiN/SiO$_2$/Ag plasmonic heterostructure at a laser excitation wavelength of $\lambda = 375$ nm as a function of applied voltage (Supplementary Fig. 13). We observe a slight decrease of the absorbance at positive voltages that implies increased excitation intensity, and consequently, increased PL intensity. The observed PL intensity modulation can also be attributed to the reduction or

increase of the absorbance of the ENZ-TiN/SiO$_2$/Ag plasmonic heterostructure at QD emission wavelengths (Supplementary Fig. 13). However, we would like to point out that the absorbance modulation at the QD emission wavelength and modulation of the total decay rate of a QD are interrelated since the QDs are placed in the immediate vicinity of the TiN layer (Fig. 1). In what follows, we further investigate how the LDOS modulation contributes to the observed PL modulation.

**Active control of quantum yield of QDs**. Once a QD has been excited by absorbing a photon, it may undergo a transition to the ground state either by far-field photon emission (radiative pathway) or via non-radiative processes, such as energy transfer via non-radiative dipole–dipole coupling[23]. An important metric to quantify the emission properties of fluorophores is the quantum yield, which is the probability of an excited QD to relax via the radiative pathway. When no electrical bias is applied, the estimated quantum yield of the InP QDs embedded in the SiO$_2$ layer of plasmonic heterostructure is 15% (Supplementary Fig. 12).

Our measurements show that under positive bias the radiative emission decay rate ($\Gamma_{rad}$) increases by 15%, while under negative bias $\Gamma_{rad}$ decreases by 11% (Fig. 5a). This amounts to a relative modulation of $\Gamma_{rad}$ of 26% when the applied gate voltage varies between −1 and +1 V (see Methods for further details). The

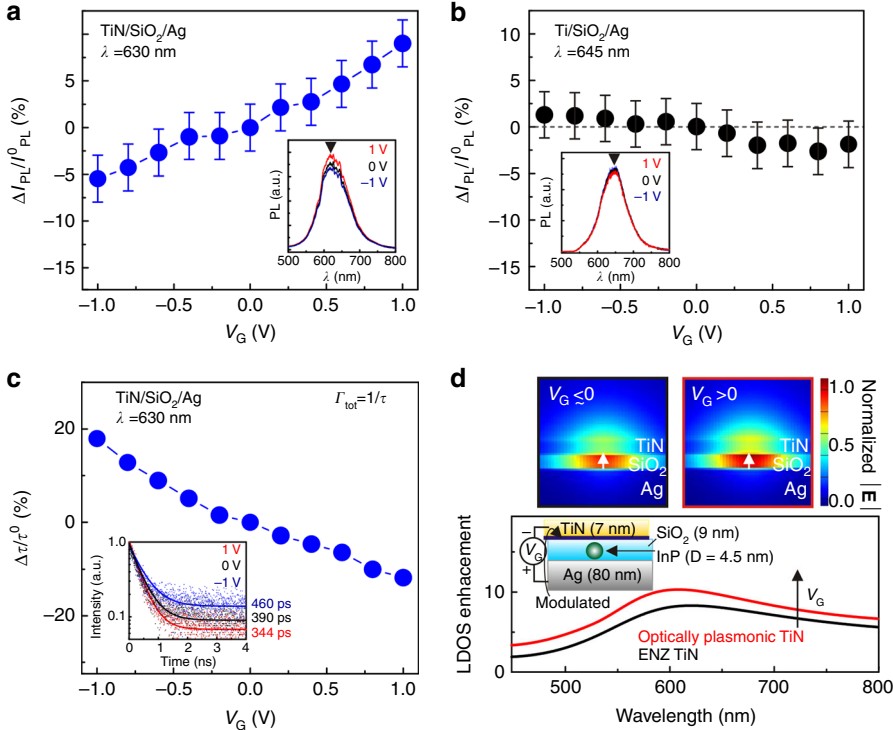

**Fig. 4** Gate-tunable spontaneous emission of quantum dots (QDs) via modulation of the local density of optical states (LDOS). **a** Modulation of the photoluminescence (PL) intensity of InP QDs embedded in the gated TiN/SiO$_2$/Ag plasmonic heterostructure for wavelength of $\lambda = 630$ nm. We observe a 10% PL relative intensity increase when the gate voltage $V_G$ is varied from 0 to +1 V. When $V_G$ is varied from 0 to −1 V, we observe a 5% PL relative intensity decrease. The inset of **a** shows the PL intensity spectra for different gate voltages. **b** PL intensity of QDs embedded in a gated Ti/SiO$_2$/Ag heterostructure for wavelength of $\lambda = 645$ nm. No modulation of PL intensity is observed under an applied bias (within the PL intensity measurement error, which is the standard deviation of the PL intensity measured at the same excitation spot of the sample at zero bias). The PL spectra for different gate voltages, plotted in the inset of **b**, also shows no modulation under applied bias. **c** PL lifetime of QDs embedded in the gated TiN/SiO$_2$/Ag plasmonic heterostructure. When the gate voltage $V_G$ is increased from 0 to +1 V, the QD lifetime decreases by 12%. When $V_G$ is varied from 0 to −1 V, the QD lifetime increases by 18%. The inset of **c** shows the PL intensity as a function of time for different gate voltages $V_G$. **d** Calculated LDOS enhancement spectra at the position of a QD (averaged over QD dipole orientations) for different carrier densities in a 1 nm-thick modulated TiN layer. The black curve corresponds a homogeneous TiN film, which is in the epsilon-near-zero (ENZ) region. The red curve corresponds to a TiN film with a 1 nm-thick modulated TiN layer that is plasmonic but far from the ENZ region. The top panels show the simulated spatial distribution of the optical frequency electric field |E| radiated by a QD ($\lambda = 630$ nm). Both the calculated LDOS and optical field intensity |E| in the SiO$_2$ gap increase with gate voltage

measured voltage-dependent total emission decay rate (Fig. 4c) and radiative decay rate can be used to determine the variation of QD quantum yields $\eta = \Gamma_{rad}/\Gamma_{tot}$ under applied bias. We observe a 35% relative increase of the quantum yield at an applied bias of +1 V and 21% relative decrease of the quantum yield at an applied bias of −1 V (Fig. 5b). This in situ control of quantum yield is a unique consequence of the bias-induced modulation of LDOS. We emphasize that the LDOS, radiative emission decay rate, and quantum yield do not depend on the absorbance at the excitation wavelength of $\lambda = 375$ nm.

To summarize, we observe that at positive bias the increase in the PL intensity is always accompanied by increases in both total and radiative emission decay rates: $\Gamma_{tot}$ and $\Gamma_{rad}$ (Figs. 4 and 5). This implies that the LDOS modulation also contributes to an increase in PL intensity under applied bias along with reduced absorption of the ENZ-TiN/SiO$_2$/Ag plasmonic heterostructure at the QD emission wavelength $\lambda = 630$ nm and slightly increased excitation intensity at $\lambda = 375$ nm. Although, as mentioned above, at the QD emission wavelength the LDOS modulation and the absorbance modulation cannot be fully decoupled. In previously reported cases of bias-induced LDOS modulation, an increase of the LDOS has always been accompanied by a decrease in the PL intensity, which implies that the applied bias primarily affects the non-radiative decay rate[13,23]. In contrast, in our work we observe a simultaneous increase in LDOS and PL intensity.

## Discussion

The ability to electrostatically control the spontaneous emission rate, intensity, and quantum yield of colloidal QDs via modulation of LDOS may lead to a number of applications, since this represents a means to dynamically modulate emitted intensity without electrical injection of carriers into the QD emitters. Thus more optimized structures featuring electrostatic LDOS modulation of optically pumped QDs have potential for application in optically efficient ultrathin displays with long durability, reduced pixel size, and large viewing angle. We note that modern commercially available QD displays typically use blue light emitting diodes (LEDs) to optically pump colloidal QDs, which down-convert the blue LED emission into red and green light[24]. The resulting QD light emission is used for backlighting, while the image is formed by the same principles as in regular liquid crystal displays[25]. Using colloidal QDs with narrowband emission at controllable wavelengths has enhanced the color gamut of modern displays, enabling generation of more realistic and vivid colors. The proof-of-concept experiment demonstrated here suggests that emission control of optically pumped QDs may be directly used for image formation, potentially eliminating necessity of the liquid crystal light modulator. Unlike the case of QD-LEDs, where both QD excitation and emission intensity modulation are achieved via current injection into the QDs, our approach decouples the excitation and modulation mechanisms.

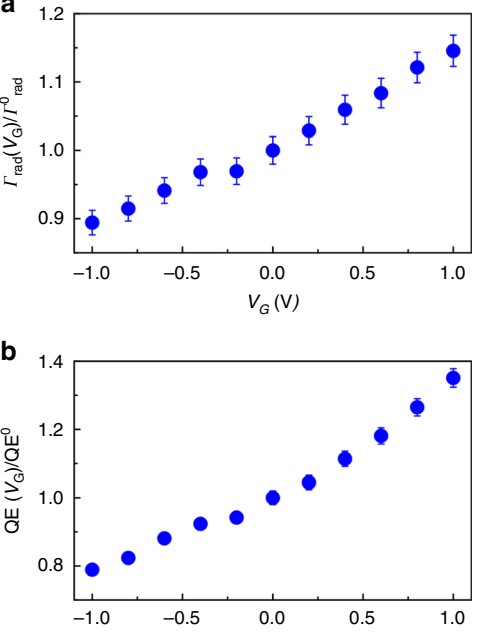

**Fig. 5** Active control of quantum yield of quantum dots (QDs) embedded in a gated plasmonic heterostructure. **a** Radiative decay rate ($\Gamma_{rad}$) of InP QDs (normalized to radiative decay rate at zero bias) embedded in the TiN/SiO$_2$/Ag plasmonic heterostructure as a function of gate voltage (within the photoluminescence (PL) intensity measurement error, which is the standard deviation of the PL intensity measured at the same excitation spot of the sample at zero bias). Under positive bias $\Gamma_{rad}$ shows a relative increase of 15% while under negative bias $\Gamma_{rad}$ shows a relative decrease of 11%. **b** Dynamically tunable quantum yield of QDs (normalized to quantum yield at zero bias). When gate voltage $V_G$ is increased from 0 to +1 V, we observe a 35% relative increase of quantum yield. As gate voltage $V_G$ varies from 0 to −1 V, we observe a relative quantum yield decrease of 21%

As a result, our method does not lead to deterioration of the emission properties of QDs typically observed in QD-LEDs. Moreover, due to capacitive nature of the device architecture used for our modulation mechanism, there is no need to place QDs between p- and n-type materials. This eliminates the necessity of the careful band alignment of the constituent materials.

When considering display applications, it is crucial to have large relative modulation of PL intensity. An obvious improvement would be to replace SiO$_2$ by a gate dielectric with higher DC permittivity, such as Al$_2$O$_3$ or HfO$_2$. The relative modulation strength of our device can further be improved by incorporating a TiN layer into a plasmonic cavity that can be formed, for instance, by placing an appropriately designed patch antenna on top of the Ag/SiO$_2$/TiN heterostructure[4,26]. Previous research has shown that the lifetime of a quantum emitter embedded in the dielectric gap of the patch antenna resonator is very sensitive to gap thickness[26]. If the resonant wavelength of the cavity is aligned with the emission wavelength of the emitter, this configuration will yield a large modulation of the PL intensity under an applied bias. Moreover, using a patch antenna resonator can increase emission intensity by orders of magnitude[4], thus making the whole device more energy efficient. The energy efficiency of our device can be further improved by using QDs with a higher quantum yield and by using a QD ensemble with narrower spectral width of PL intensity (for a more detailed discussion, see Supplementary Note 7).

Our findings can be readily used for visible light communication systems such as Li-Fi, a counterpart of Wi-Fi that utilizes the visible spectral range[27,28]. Li-Fi denotes the proposed future visible light communication scheme, in which temporally modulated LED lamps are likely candidates as sources for an optical wireless communication system to transmit information encoded via intensity variations in the emitted light. Cadmium-free QDs are currently used for improving the quality of light of LED light bulbs by making the emitted light more pleasant for the human eye. In many lighting devices, QDs absorb blue LED light and re-emitted as light downshifted in wavelength to achieve a desired lamp emission color characteristic. It has been suggested that light intensity modulations could be achieved by varying the intensity of the blue LED pump[27,28]. Instead of varying the intensity of the LED pump light source, we propose to dynamically control the emission properties of QDs via LDOS modulation. Our proposed approach would enable the realization of a more flexible visible light communication system, where, for example, an actively tunable wavelength multiplexing device could be enabled by separately modulating the emission intensity of QDs with different emission wavelengths. We would like emphasize that, within the context of visible light communication systems, a large amplitude modulation of the emitted light is not necessary[27,28].

## Methods

**Sample fabrication.** We formed a dilute solution of commercially available InP/ZnS core–shell QDs (2 mgmL$^{-1}$ in toluene) in isopropanol, with a volume ratio of 1:20 and a quantum yield of 17%. Next, we fabricated TiN/SiO$_2$/Ag plasmonic heterostructures with QDs embedded in the SiO$_2$ layer by depositing 80 nm of Ag, followed by 4 nm of SiO$_2$ on a mica substrate via electron-beam evaporation at a chamber pressure of 10$^{-7}$ torr. We then spin-coated the diluted QD solution for 30 s at 4000 rpm to spread the solution evenly and subsequently deposited another 4 nm of SiO$_2$ via electron-beam evaporation. In this way, we obtained a layer of QDs embedded in the SiO$_2$ layer. As a final step, we sputtered 7 nm TiN film with DC magnetron sputtering at a chamber pressure of $7.5 \times 10^{-7}$ torr and at room temperature. In accordance with previously reported results[16–18], varying argon/nitrogen flow rate, DC power, growth temperature, and growth substrate strongly influenced the optical properties of the deposited films. Using Hall measurements, we identified the growth conditions enabling the carrier concentration of the fabricated TiN films to lie between $N = 5.9 \times 10^{20}$ cm$^{-3}$ and $N = 4.1 \times 10^{22}$ cm$^{-3}$, while the mobility ranged from 0.059 to 5.8 cm$^2$ V$^{-1}$ s$^{-1}$ (Supplementary Tables 1 and 2). Each deposition step was undertaken with an appropriate face mask made of stainless steel to permit easy bias application configuration in the final device. The working area of the device was $1 \times 1$ mm$^2$. The thicknesses of the constituent layers in the fabricated TiN/SiO$_2$/Ag heterostructure were determined via cross-sectional transmission electron microscopic analysis. To prepare the sample for the electron microscopic analysis, we used dual-beam focused gallium ion-beam (FIB, FEI model Nova 200) to FIB-cut the heterostructure normal to the interface. The diameter of the QDs was 4–5 nm, as determined by high-resolution transmission electron microscopy (FEI Tecnai TF20 STEM). To determine the material parameters of the sputtered TiN films, we fabricated additional large-area control samples that consisted of 35 nm of SiO$_2$ and about 30 nm of TiN deposited on Si substrate under the same growth conditions as the ones used to fabricate our final devices. We used large-area control samples for both Hall and ellipsometry (J.A. Woollam Co. model VASE) measurements. Since the optical properties of TiN film sensitively depend on the choice of the substrate[19], our control samples used for both ellipsometry and Hall measurements were deposited on SiO$_2$.

**Optical measurements.** Reflectance spectra were taken under normal incidence with an objective 5× (Olympus, with a numerical aperture of 0.14) focusing a supercontinuum laser (Fianium) down to a small spot of 3 μm in diameter. The reflected signal was measured by a silicon photodetector. The micro-PL and time-resolved PL measurements were taken on an inverted microscope (Zeiss, Inc.) equipped with a spectrometer consisting of a monochromator and a liquid-nitrogen-cooled CCD camera. For PL lifetime measurements, a 375 nm picosecond laser diode (70 ps pulse duration, 40 MHz repetition rate; PicoQuant) excitation source was used, and a 375 nm band pass filter was placed after laser source to purify the laser beam. A 100× objective lens with a numerical aperture of 0.9 (Zeiss, Inc.) was used to focus the pulsed laser to a small spot of $1.6 \times 10^{-6}$ cm$^2$ with an estimated peak power density of 7.5 kW cm$^{-2}$. We measured QD lifetime by using a time-correlated single photon counting module (PicoHarp 300, PicoQuant) and single photon avalanche diode detector (PDM 50 T, MicroPhoton Devices)[29]. We used periodic pulsed laser excitation and collect photons from multiple excitation and emission cycles. This approach allowed us to construct a histogram of number of photon counts at different photon arrival times. The time resolution of our lifetime measurement setup was 200 ps. PL lifetime decays were acquired from a

particular region <5 µm in diameter that was illuminated with an iris. During lifetime measurements, a 600–650 nm band-pass filter was used. When measuring the QD lifetime, we illuminated the same region of the sample that was used for PL measurements. This enabled direct comparison of the QD lifetime and PL intensity data sets and further analysis of the radiative and non-radiative recombination rate. Even though we measured slight differences in the emission spectrum depending on the part of the sample that was illuminated, we obtained reproducible data when the position of the illumination spot was fixed. We also verified that the shape of a PL intensity spectrum does not depend on pump field intensity (Supplementary Fig. 16). In addition, the voltage bias was applied to the heterostructure by a source meter (Keithley 2400 source-meter).

**Electromagnetic simulations**. Numerical electromagnetic simulations were performed using a finite difference time domain method. When modeling the performance of our structure under applied bias, we assumed that the complex refractive index of a 1 nm-thick TiN layer at the interface with $SiO_2$ is modulated by the applied bias (Supplementary Figs. 6, 9 and 11).

**Quantum yield measurements**. To measure the quantum yield of our InP QDs, we used a standard florescence dye (Rhodamine 6G) with a known quantum yield of 95% (Supplementary Fig. 12). When calculating radiative emission decay rate, we took into account that our InP QDs were embedded in the $SiO_2$ layer of the ENZ-TiN/$SiO_2$/Ag heterostructure. Since the QDs were embedded in the ENZ-TiN/$SiO_2$/Ag heterostructure, a portion of the laser excitation ($\lambda = 375$ nm) was absorbed in the top TiN layer, which affected the excitation intensity of the QDs. To estimate the effect of the possible variation of the excitation intensity, we measured the absorbance of the ENZ-TiN/$SiO_2$/Ag plasmonic heterostructure at an excitation wavelength of $\lambda = 375$ nm under an applied bias. As seen in Supplementary Figs. 13 and 14, the absorbance stayed almost constant for negative biases and showed a slight decrease for positive biases. Since absorption primarily occurred in the TiN layer, high absorbance resulted in the reduced excitation intensity of the QDs. Taking this into account, the bias dependent radiative emission decay rate ($\Gamma_{rad}$) given by the following formula[26,30]:

$$\Gamma_{rad}(V_G)/\Gamma_{rad}^0 = \left(I_{PL}(V_G)/I_{PL}^0\right)\left[(1 - A_{laser}(V_G))/\left(1 - A_{laser}^0\right)\right]. \quad (1)$$

Here $\Gamma_{rad}^0$ the radiative emission decay rate under zero bias, $I_{PL}^0$ the peak PL intensity under zero bias, $I_{PL}(V_G)$ the bias-dependent PL intensity, $A_{laser}^0$ the absorbance in the ENZ-TiN/$SiO_2$/Ag heterostructure at an excitation wavelength of $\lambda = 375$ nm at zero bias, and $A_{laser}(V_G)$ the bias dependent absorbance at $\lambda = 375$ nm. $V_G$ denotes the gate voltage applied between the Ag and TiN layers in the TiN/$SiO_2$/Ag heterostructure. We would like to emphasize that absorbance in the TiN/$SiO_2$/Ag heterostructure primarily occurred in TiN layer. To calculate the bias-dependent quantum yield of our QDs embedded in TiN/$SiO_2$/Ag heterostructure, we used Eq. (1) and took into account that quantum yield of the emitter $\eta$ was defined as a ratio of radiative and total decay rates $\eta = \Gamma_{rad}/\Gamma_{tot}$.

**Data availability**. The data sets analyzed during this study are available from the corresponding author on reasonable request.

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

## Acknowledgements

This work was supported by Samsung Electronics, the Air Force Office of Scientific Research under grant number FA9550-16-1-0019 (device-related work), and the Department of Energy under grant number DE-FG02-07ER46405 (transparent conductor-related work). We also acknowledge use of facilities supported by the Kavli Nanoscience Institute (KNI) and the Joint Center for Artificial Photosynthesis (JCAP) at Caltech. Y.-J.L. acknowledges the support from Ministry of Science and Technology, Taiwan (Grant numbers: 104-2917-I-564-057). The authors would like to thank Wei-Hsiang Lin, Anya Mitskovets, and Panos Patsalas for useful discussions. The authors gratefully acknowledge Erin Burkett from the Hixon Writing Center at Caltech for providing feedback and guidance on writing during the revision process. The authors also deeply appreciate help in the form of the close reading of the manuscript and review responses by Kelly Mauser, Dagny Fleischman, Rebecca Glaudell, Haley Bauser, Cora Went, Phil Jahelka, and Michael Kelzenberg.

## Author contributions

Y.-J.L., R.S. and H.A.A. proposed the original idea. Y.-J.L. performed all experiments, calculations, and data analysis. R.S. proposed the theoretical model and performed calculations. R.P. and W.-H.C. helped with the optical setup. K.T., G.K.S., W.-H.C., and R.S. performed ellipsometry measurements and analyzed the ellipsometry data. A.R.D. helped in discussion. Y.-J.L., R.S., and H.A.A. wrote the paper, and all authors discussed and revised the final manuscript.

## Additional information

**Competing interests:** The authors declare no competing financial interests.

