## [Peer Review File · Nature Communications]

Reviewers' comments:

Reviewer #1 (Remarks to the Author):

The authors report on an experimental work on dynamic modulation of spontaneous emission from quantum emitters in a plasmonic heterostructure. The general idea is to introduce quantum dots in a gated metal-insulator-semiconductor device whose epsilon-near-zero (ENZ) dispersion regime can be tuned through the injection or removal of free carriers. By electrostatically gating the device, the local density of optical states (LDOS) felt by the emitters in the cavity is modified, hence dynamically modulating the emission into the far-field. The overall idea is interesting and original. This naturally follows on from the authors previous works on gate tunable ITO [Nano letters 10, 2111 (2010), and ref. 9 in the manuscript]. However, as shown in Fig. 4a -- which is the main direct result of the paper -- the reported modulation amplitude is somewhat weak. Indeed, the maximum modulation depth is less than 15% when the applied voltage is varied from -1V to +1V. It is hard to envision how such a low modulation amplitude could be used in applications such as the one mentioned by the authors (displays). Furthermore, there are no indications on a potential improvement of this modulation amplitude. For this reason, I have doubts on the future impact of such a work. In addition, I have a number of comments concerning the manuscript:

i) The Fig. 4a shows a modulation of the PL intensity as detected in the far field, which is the main direct result of the study. This change in PL intensity is not an evidence per se of a "Purcell modulation", as this result could arise from multiple other factors such as a change in the non-radiative decay rate, a modification in the absorption or even a change in the pump efficiency. For example, one could alternatively explain the increase (decrease) in PL intensity when $V_g > 0$ ($V_g < 0$) to be produced by a decrease (increase) in the absorbance of the device (displayed in Fig. S9). However, the authors measured the quantum yield of the emitters and its variation as a function of the applied voltage. From that they were able to extract the variation of the radiative rate (even though it is not explicitly said in the paper, I assume they used the measured total lifetime to estimate the radiative rate following the relation $\Gamma_{rad} = QY * \Gamma_{tot}$). The corresponding variation of radiative rates with the applied voltage is therefore a proof on the enhancement of the LDOS. This also indicates that all of the claims in the paper rely on an accurate measurement of the quantum yield. However, even though there is a substantial description on how the quantum yield experiment was performed, there is only one graph on the related results of quantum yield measurements, corresponding to $V_g = 0$ (Fig. S8). The authors should display all of their data related to quantum yield measurement as a function of V_g in the supplementary. For the same reasons, they should display the error bars in both the quantum yield measurements (Fig. 5) and the lifetime measurements (Fig. 4c). (they have done so for the PL measurements so they should stay consistent throughout the paper).

ii) The LDOS enhancement, as calculated and displayed in Fig. 4d is only valid for a z-oriented dipole. It is mentioned in the supplementary that a qualitatively similar result would be obtained in a more realistic case where dipoles are isotropically oriented. I believe it would be worth showing the corresponding LDOS enhancement for isotropic emitters as a function of wavelength, so that one will be able to compare this result to the values obtained experimentally in the paper. In other words, does the simulation indicate that we should effectively see a twofold enhancement in the radiative rate upon modulating the bias from -1V to +1V? Or is there any other competing effects at play?

iii) The evolution of the imaginary part of permittivity against the carrier concentration, shown in Fig. 2 is rather strange. Indeed, the $\text{Im}(\epsilon)$ constantly increases with the concentration of free carriers but suddenly decreases between $N = 1.8 * 10^{22}$ and $4.1 * 10^{22} \text{ cm}^{-3}$. In addition, for this latter sample, the evolution of $\text{Im}(\epsilon)$ as a function of wavelength is not monotonic, what is also surprising if we compare to other studies. And finally, the fitting model for this sample is different: the authors

used 3 Lorentz oscillators in that particular sample while they used 2 Lorentz oscillators for all other cases. This does not necessarily mean that the fit is wrong, but it is a strong indication that this particular sample is different than the others (probably due to structural differences between samples). Therefore, I strongly doubt a TiN thin film, initially prepared as ENZ-TiN, will effectively see its optical dispersion behaves similarly as in Fig. 2 upon injection of carriers through electrode gating. This is corroborated by results shown in Fig. S2: if the accumulation (depletion) of carriers in ENZ-TiN was qualitatively similar to changing the TiN from optically plasmonic to optically dielectric, the reflectance value of ENZ-TiN should oscillate between the respective ones of plasmonic and dielectric TiN (80% for optically plasmonic, 95% for optically dielectric, as displayed in Fig. S2). This is obviously not the case here, as the reflectance of ENZ-TiN under applied bias changes from ~67% to ~82%, what indicates that the effective changes in the optical dispersion of ENZ-TiN are different than the optical dispersions shown in Fig.2. Therefore, I believe a better way of presenting things would be to show the optical modulation against the gate voltage as measured by ellipsometry (similarly to what was done by the authors in Nano letters 10 2111 (2010)).

iv) There are 4 TiN samples displayed in Fig. 2 and the corresponding material properties values of all 4 are shown in table S1, but in the Drude-Lorentz fitting parameters table S2, only 3 of them are shown. Why? This is especially surprising given that the missing sample is the most important one, i.e. the ENZ-TiN that is used for all of the modulation devices. This data needs to be included in the supplementary.

v) The dynamic modulation is convincingly shown in Fig. S3 but there are no indications on the modulation amplitude. It should be explicitly mentioned in the paper. Also, why is the applied bias $\pm 5V$ in that case while it is $\pm 1V$ throughout the manuscript? What happens in the device containing QDs when the voltage is increased to voltages higher than 1V. The authors should explain that in the manuscript as it could be a limitation to potential applications.

In conclusion, this manuscript starts on an interesting concept and shows some first proof-of-principle but I believe the different points I raised should be addressed before considering this paper for publication in Nature Communications. Especially, the authors should give details on how the modulation amplitude of their device could be improved.

Reviewer #2 (Remarks to the Author):

In this manuscript, the authors experimentally demonstrated dynamic control of the visible spontaneous emission of colloidal quantum dots by electrically tuning the local optical environment. Specifically, by carefully changing the doping level of TiN, its permittivity can reach near zero (ENZ) at the visible wavelength regime. Using external voltage to build up a charge depletion or accumulation layer in TiN, its refractive index will be greatly tuned near ENZ wavelength, thus leading to the LDOS change of QDs and resulting in the modulation of spontaneous emission.

This manuscript is well written and the results are technologically sound. Electrically active control of the visible light emission have great potential applications. The layered structure in this work can be easily fabricated and the dynamic modulation through LDOS could reach ultrafast speed. I believe it is a good contribution to the field of optoelectronics. Thus, I recommend it to be considered for publication in Nature Communications after addressing the following concerns.

1. While discussing the photoluminescence (PL) intensity modulation (i.e., lines 88-99 of page 4), the authors attributed the modulation to LDOS change caused by varied external voltages (Fig. 4a). However, although the external optical pump power was constant for varied voltages, the actual pump

field intensity at the emitter locations might still vary for different voltages, thus leading to modulation of PL intensity as well. Although the lifetime results (based on LDOS) in Fig. 4c can partially explain the modulated PL intensity, the pump effect may also contribute to such modulation. As shown in Fig. 3a, the reflectance varies (in wide spectral range ~450 to 800nm) for different external voltages. The authors should comment on this pump effect of 375nm excitation wavelength in the main text.

2. In the section "Active control of quantum yield of QDs" (lines 109-123), the authors only briefly summarized the results of radiative/non-radiative rate and quantum yields. A little more discussion about the observed results may be helpful for the readers. In addition, although the experimental details for this section are covered in the Supplementary Materials, it may be better to add some descriptions in either the main text or the Methods section.

Reviewer #3 (Remarks to the Author):

The manuscript presented by Lu et al reports the demonstration of an interesting and long-sought-after mechanism to control the spontaneous emission yield of InP / ZnS core-shell nanostructures by dynamically controlling the local density of photonic modes they experience using an electrically tunable nano-plasmonic device. The use of the TiN layer as a tunable plasmonic material, having a plasma frequency in the visible range, is an interesting result that will certainly be of interest to the community of researchers working in photonics and plasmonics. However, the observed effect is weak, consisting of a $\pm 7\%$ change in the radiative luminescence efficiency as the voltage applied to the field-effect capacitor is tuned from -1V to +1V. Moreover, whilst an impressive array of supplementary material is presented that certainly helps to support the conclusions that the spontaneous emission rate, and hence the quantum efficiency, of the colloidal QDs is indeed varied by tuning the electric field, but I have some questions for the authors that should be addressed before publication is further considered. I feel that the topic of the paper is of sufficient interest and timeliness to warrant publication in nature communications, providing that the technical concerns raised below are fully addressed.

- The predictions of fig 4d would indicate that one might expect a tunable enhancement of the LDOS over the spectral range between ca 550-750nm, with a maximum response occurring around ~600nm, but active over the whole of this spectral range. In the PL spectra presented in the inset of fig 4a, the field induced emission enhancement seems to be only present in a much narrower spectral range 600-660nm. How do the authors account for this discrepancy?
- There does seem to be a shift of the peak position of the PL data presented in the inset of fig 4a upon modifying the applied voltage. Large static electric fields can impact on the average charge status of the quantum dots and such effects can give rise to both spectral shifts and a change in the radiative efficiency. How can the authors discount the possibility that the charge status of the dots are tuned by the electric field, giving rise to the observed change in radiative efficiency? Here, perhaps it would help to present the raw PL data more prominently (larger panel, logarithmic scale, differential spectra recorded with a gate voltage V , relative to the spectra obtained at $V=0$...). This is to my mind a crucial point, since the spectral dependence of the LDOS modulation / enhancement for ENZ TiN and Optically Plasmonic TiN is likely to be of significant interest to readers and the result should be unambiguous.
- The use of either two or three Lorentz oscillators to fit the ellipsometry data and produce the carrier density dependent dielectric function (fig 2) seems to be somewhat arbitrary? Could the authors please relate the frequencies of the Lorentz oscillators to the expected bandstructure of the n-doped TiN and explain the rationale behind the choice of using either 2 or 3 Lorentz oscillators to fit the

dielectric function as the free carrier density N varies?

- The method used to record the time resolved data involves integrating over the 500-650nm spectral range. This approach is reasonable, but assumes that the form of the emission spectrum is independent of the excitation level. Did the authors check that the form of the emission spectrum was not time dependent ?

Detailed Response to Reviewer #1's Comments:

The authors report on an experimental work on dynamic modulation of spontaneous emission from quantum emitters in a plasmonic heterostructure. The general idea is to introduce quantum dots in a gated metal-insulator-semiconductor device whose epsilon-near-zero (ENZ) dispersion regime can be tuned through the injection or removal of free carriers. By electrostatically gating the device, the local density of optical states (LDOS) felt by the emitters in the cavity is modified, hence dynamically modulating the emission into the far-field. The overall idea is interesting and original. This naturally follows on from the authors previous works on gate tunable ITO [Nano letters 10, 2111 (2010), and ref. 9 in the manuscript]. However, as shown in Fig. 4a -- which is the main direct result of the paper -- the reported modulation amplitude is somewhat weak. Indeed, the maximum modulation depth is less than 15% when the applied voltage is varied from -1V to +1V. It is hard to envision how such a low modulation amplitude could be used in applications such as the one mentioned by the authors (displays). Furthermore, there are no indications on a potential improvement of this modulation amplitude. For this reason, I have doubts on the future impact of such a work. In addition, I have a number of comments concerning the manuscript:

In conclusion, this manuscript starts on an interesting concept and shows some first proof-of-principle but I believe the different points I raised should be addressed before considering this paper for publication in Nature Communications. Especially, the authors should give details on how the modulation amplitude of their device could be improved.

We thank the reviewer for her or his insightful comments. First, we would like to mention that besides the proposed display applications, our findings could be used for visible light communication systems such as Li-Fi, a counterpart to Wi-Fi that utilizes the visible range of the spectrum [1, 2]. In future visible light communication networks, LED lights are likely candidates for access points that will transmit information via subtle intensity variations encoded in the emitted light. It is notable that cadmium-free QDs are currently used for improving the quality of light from LED light bulbs. In such lighting devices, the QDs absorb blue LED light, down-convert it, and re-emit it with the desired color characteristics. In principle, instead of modulating the intensity of the LED pump, one could modulate the emission properties of QDs via LDOS modulation. The proposed approach would enable more flexible visible light communication systems. For example, by separately modulating the emission intensity of QDs with different emission wavelengths, one could obtain an actively tunable wavelength multiplexing device. Notably, within the context of visible light communication systems, one does not need to have large modulation depth of the emitted light [1, 2].

When considering display applications, indeed, having large modulation depth is very important. We have some suggestions how we could increase modulation amplitude of our device:

a. Using gate dielectrics with larger DC permittivity

In our design we use SiO₂ as a gate dielectric since it is known to have very low leakage current. However, the DC permittivity of SiO₂ is also relatively low ($\epsilon_{dc}=4.5$). One straightforward improvement could be to replace the SiO₂ with a dielectric having higher DC permittivity, such as HfO₂ ($\epsilon_{dc}=25$) [3]. This would enable larger variation of the carrier concentration in the TiN and, hence, larger LDOS and PL intensity modulation.

b. Using resonant structures

We suggest using a plasmonic cavity (Fig. S9a) that can be formed by placing an appropriately designed patch antenna on top of the Ag/SiO₂/TiN heterostructure (akin to structure shown in Refs. [4, 5]). It has been shown that the lifetime of a quantum emitter embedded in the dielectric gap of the patch antenna resonator is very sensitive to the gap thickness [5]. If the resonant wavelength of the cavity is aligned with the emission wavelength of the emitter, this configuration will yield large modulation of the PL intensity under applied bias. Figure S9b shows the distribution of the electric field excited by the point dipole embedded in a 5 nm-thick gap of the proposed patch antenna structure (Fig. S9a). The assumed dielectric permittivity of the 7 nm-thick TiN film is identical to that of the ENZ-TiN film used in the present work (i.e., the film with carrier concentration $N=1.8\times 10^{22}$ cm⁻³ from Fig. 2). In case of positive bias, we assume that a 1 nm-thick accumulation layer is formed in the TiN, at the SiO₂ interface. The dielectric permittivity of the accumulation layer is assumed to be identical to that of the metallic TiN described in the present work (i.e., the film with carrier concentration $N=4.1\times 10^{22}$ cm⁻³ from Fig. 2). Figure S9c shows the LDOS enhancement for two different biases. As one can see, the suggested structure enables significant LDOS modulation under applied bias. Even though the dielectric permittivity of the accumulation layer of TiN is likely to differ from that of metallic TiN as assumed in the present work, the simulation shown in Fig. S9 aims to demonstrate the sensitivity of the lifetime of the emitter with respect to variation of the refractive index of the material in the gap. Finally, we would like to note that using patch antenna resonators can increase the emission intensity from the device by orders of magnitude [4].

Figure S9 | (a) Schematic of the proposed patch antenna with a quantum emitter embedded in a 5 nm-thick SiO₂ layer. (b) Calculated electric field distribution in the patch antenna structure. The depicted electric field is excited by the point dipole oriented normally to the planar layers of Ag, SiO₂, and TiN. (c) Calculated z-projection of the LDOS enhancement with respect to bulk InP LDOS, as a function of the emission wavelength, for zero bias and positive bias.

Finally, we would like to mention that the energy efficiency of the proposed device can be considerably improved by using higher quality QDs:

c. Using quantum dots with higher quantum yield

The quantum yield of our InP/ZnS QDs suspended in the solution is 17%, while the measured quantum yield of QDs embedded in a 9 nm-thick SiO₂ layer is 15%. This modest value of quantum yield implies that only 15% of the photons absorbed by the QDs will be reradiated into the far field. Hence, using QDs with higher quantum yield would enhance emission intensity from the structure, and hence improve the absolute (but not relative) modulation of PL intensity under applied bias.

d. Using a quantum dot ensemble with narrower spectral width of PL intensity.

Note that the experimentally observed PL intensity spectrum is formed by emission of multiple colloidal QDs with different sizes and, consequently, different emission wavelengths. The size distribution of QDs gives rise to a broad PL intensity spectrum observed in our experiment (see Fig. 3). The observed PL intensity spectrum is centered at 630 nm and its full width at half maximum

(FWHM) is around 90 nm. Since the LDOS in the SiO₂ spacer of the planar heterostructure has no resonant features (Fig. 4d), we would expect a fairly broadband modulation of PL intensity. Thus, in principle, PL intensity is equally modulated for each individual QD coupled to the plasmonic heterostructure. However, due to larger number of QDs emitting in the wavelength range from 600 nm to 650 nm, absolute value of the PL intensity modulation (but not relative PL intensity change) is larger for the central emission wavelength. If, for an identical QD density, the considered QD ensemble had a narrower PL intensity spectrum, then the absolute, but not relative, modulation at the central emission wavelength would be stronger due to the large number of QDs emitting at a single wavelength. Note that some commercially available QDs have 30-40 nm FWHM over whole visible spectrum [6]. Thus, using these QDs in our device would significantly enhance absolute value of the PL intensity modulation at a given intensity of pump field.

We added the following paragraphs to the main text of the manuscript:

“When considering display applications, it is crucial to have large relative modulation of PL intensity. An obvious improvement would be replacing SiO₂ by a gate dielectric with higher DC permittivity such as Al₂O₃ or HfO₂. The relative modulation strength of our device can further be improved by incorporating a TiN layer into a plasmonic cavity that can be formed, e.g., by placing an appropriately designed patch antenna on top of the Ag/SiO₂/TiN heterostructure [4, 5]. It has been shown that the lifetime of a quantum emitter embedded in the dielectric gap of the patch antenna resonator is very sensitive to gap thickness [5]. If the resonant wavelength of the cavity is aligned with the emission wavelength of the emitter, this configuration will yield a large modulation of PL intensity under an applied bias (see SI). Moreover, using a patch antenna resonator can increase emission intensity by orders of magnitude [4] thus making the whole device more energy efficient. The energy efficiency of our device can further be improved by using QDs with a higher quantum yield, and by using a QD ensemble with narrower spectral width of PL intensity (for a more detailed discussion see SI).

Our findings can readily be used for visible light communication systems such as Li-Fi, a counterpart of Wi-Fi that utilizes the visible spectral range [1, 2]. In future visible light communication networks, LED light bulbs are likely candidates for devices that will transmit information encoded via subtle intensity variations of the emitted light. Cadmium-free QDs are currently used for improving quality of light of LED light bulbs by making the emitted light more pleasant for the human eye. In many lighting devices QDs absorb blue LED light, and then down-convert and re-emit it with the desired color characteristics. It is currently suggested that light intensity modulations can be achieved by varying the intensity of the blue LED pump. Instead of varying intensity of LED pump, we propose to dynamically

control emission properties of QDs via LDOS modulation. The proposed approach would enable realizing more flexible visible light communication systems. For example, an actively tunable wavelength multiplexing device can be obtained by separately modulating the emission intensity of QDs with different emission wavelengths. We would like emphasize that within the context of visible light communication systems, a large amplitude modulation of the emitted light is not necessary [1, 2].”

We also added the following part to the Supplementary Information (SI):

“To enhance the amount of observed LDOS modulation, we suggest using a plasmonic cavity (Fig. S9a) that can be formed by placing an appropriately designed patch antenna on top of an Ag/SiO₂/TiN heterostructure (akin to previously demonstrated reflectarray antennas [4, 5]). It has been shown that the lifetime of a quantum emitter embedded in the dielectric gap of the patch antenna resonator is very sensitive to gap thickness [5]. If the resonant wavelength of the cavity is aligned with the emission wavelength of the emitter, this configuration will yield a large modulation of the PL intensity under an applied bias. Figure S9b shows the distribution of the electric field excited by the point dipole embedded in the 5 nm thick gap of the proposed patch antenna structure (Fig. S9a). The assumed dielectric permittivity of 7 nm thick TiN film is identical to that of ENZ-TiN film used in the present work (the film with carrier concentration $N= 1.8 \times 10^{22} \text{ cm}^{-3}$ from Fig. 2). In the case of a positive bias, we assume that a 1 nm thick accumulation layer is formed in TiN at the TiN/SiO₂ interface. The dielectric permittivity of the accumulation layer is assumed to be identical to that of the metallic TiN described in the present work (the film with carrier concentration $N= 4.1 \times 10^{22} \text{ cm}^{-3}$ from Fig. 2). The LDOS enhancement for two different biases is shown in Figure S9c, where it can be seen that the suggested structure enables significant LDOS modulation under an applied bias. The simulation shown in Fig. S9 aims to demonstrate the sensitivity of the lifetime of the emitter with respect to variation of the refractive index of the material in the gap, despite the difference in dielectric permittivity between the value assumed for the TiN accumulation layer and that of metallic TiN derived in the present work ($N= 4.1 \times 10^{22} \text{ cm}^{-3}$). Finally, we would like to note that using patch antenna resonator can increase the emission intensity from the device by two orders of magnitude [4].”

Figure S9 | (a) Schematic of the proposed patch antenna with a quantum emitter embedded in 5 nm thick SiO₂ layer. (b) Calculated electric field distribution in the patch antenna structure. The depicted electric field is excited by the point dipole oriented normal to the planar layers of Ag, SiO₂, and TiN. (c) Calculated z-projection of the LDOS enhancement with respect to LDOS in the bulk InP as a function of the emission wavelength for both zero bias and positive bias.”

R1-1. The Fig. 4a shows a modulation of the PL intensity as detected in the far field, which is the main direct result of the study. This change in PL intensity is not an evidence per se of a “Purcell modulation”, as this result could arise from multiple other factors such as a change in the non-radiative decay rate, a modification in the absorption or even a change in the pump efficiency. For example, one could alternatively explain the increase (decrease) in PL intensity when $V_g > 0$ ($V_g < 0$) to be produced by a decrease (increase) in the absorbance of the device (displayed in Fig. S9). However, the authors measured the quantum yield of the emitters and its variation as a function of the applied voltage. From that they were able to extract the variation of the radiative rate (even though it is not explicitly said in the paper, I assume they used the measured total lifetime to estimate the radiative rate following the relation $\Gamma_{rad} = QY * \Gamma_{tot}$). The corresponding variation of radiative rates with the applied voltage is therefore a proof on the enhancement of the LDOS. This also indicates that all of the claims in the paper rely on an accurate measurement of the quantum yield. However, even though there is a substantial description on how the quantum yield experiment was performed, there is only one graph on the related results of quantum yield measurements, corresponding to $V_g = 0$ (Fig. S8). The authors should display all of their data related to quantum yield measurement as a function of V_g in the supplementary. For the same

reasons, they should display the error bars in both the quantum yield measurements (Fig. 5) and the lifetime measurements (Fig. 4c). (they have done so for the PL measurements so they should stay consistent throughout the paper).

We agree with the reviewer's observation that a change in PL intensity is not direct evidence of LDOS modulation or "Purcell modulation". However, modulation of the emitter lifetime (or, equivalently, total decay rate Γ_{tot}) is direct evidence of LDOS modulation since Γ_{tot} is proportional to LDOS [7]. Increasing both non-radiative and radiative decay rates, Γ_{nr} and Γ_{rad} , respectively, may contribute to LDOS enhancement since ($\Gamma_{tot} = \Gamma_{rad} + \Gamma_{nr}$). Thus, in principle, LDOS enhancement can also be accompanied by a decrease in PL intensity [8, 9]. Hence, to prove that LDOS enhancement contributes to the observed increase in PL intensity, we study the variation of both total decay rate Γ_{tot} and radiative emission decay rate Γ_{rad} under an applied bias. From this we can deduce variation of quantum yield of our QDs under an applied bias. As Reviewer 1 mentioned, in our work we define quantum yield of an emitter η as a ratio of radiative and total decay rates $\eta = \Gamma_{rad} / \Gamma_{tot}$.

Following the reviewer's suggestion, we critically analyzed our approach to calculate the radiative emission rate and quantum yield of QDs. Below is the summary of the approach we used that has also been added to the Methods section of the manuscript. When calculating the radiative emission decay rate, we take into account that our InP QDs are embedded in the SiO₂ layer of the TiN/SiO₂/Ag heterostructure. As a result, a portion of the laser excitation ($\lambda=375$ nm) is going to be absorbed in the top TiN layer, which will affect the excitation intensity of the QDs. To estimate the effect of the possible variation of the excitation intensity, we measure the absorbance of the ENZ-TiN/SiO₂/Ag plasmonic heterostructure at an excitation wavelength of $\lambda=375$ nm under an applied bias (Figs. S13 and S14). As can be seen, the absorbance stays almost constant for negative biases and shows a slight decrease for positive biases. Since absorption primarily occurs in the TiN layer, a high absorbance results in a reduced excitation intensity of the QDs. Taking this into account, the bias dependent radiative emission decay rate (Γ_{rad}) can be given by the following formula [5, 10]:

$$\Gamma_{rad}(V) / \Gamma_{rad}^0 = (I_{PL}(V) / I_{PL}^0) [(1 - A_{laser}(V)) / (1 - A_{laser}^0)]. \quad (1)$$

Here Γ_{rad}^0 is the radiative emission decay rate under zero bias, I_{PL}^0 is the peak PL intensity under zero bias, $I_{PL}(V)$ is the bias-dependent PL intensity, A_{laser}^0 is the absorbance in TiN/SiO₂/Ag heterostructure at excitation wavelength of $\lambda=375$ nm at zero bias, and $A_{laser}(V)$ is the bias dependent absorbance at $\lambda=375$ nm. We would like to emphasize that absorbance in TiN/SiO₂/Ag heterostructure primarily occurs in TiN layer. To calculate the bias-dependent quantum yield of our QDs embedded in TiN/SiO₂/Ag

heterostructure, we use Eq. (1) and take into account that quantum yield of the emitter η is defined as a ratio of radiative and total decay rates $\eta = \Gamma_{\text{rad}}/\Gamma_{\text{tot}}$.

To address the reviewer's concerns regarding the relationship between LDOS and measured PL intensity, we added the following two paragraphs to the main text of the manuscript.

“Thus, in our experiment we observe simultaneous increase (decrease) of PL intensity and total decay rate of emission Γ_{tot} . However, it still needs to be proven that the measured LDOS modulation contributes to the observed PL intensity modulation. For example, variation of excitation field intensity under applied bias could also result in modulation of PL intensity. To investigate this option, we measure the absorbance spectrum of the ENZ-TiN/SiO₂/Ag plasmonic heterostructure at the laser excitation wavelength of $\lambda=375$ nm (Fig. S13a) as a function of applied voltage. We observe a slight decrease in absorbance at positive voltages, which implies an increased excitation intensity, and consequently, an increased PL intensity. The observed PL intensity modulation can also be attributed to the reduction or increase of absorbance of the ENZ-TiN/SiO₂/Ag plasmonic heterostructure at the QD emission wavelengths (Fig. S13b). However, we would like to point out that absorbance modulation at the QD emission wavelength and modulation of the total decay rate of a QD are interrelated since QDs are placed in the immediate vicinity of TiN layer (see Fig. 1). In what follows we further investigate how LDOS modulation contributes to the observed PL modulation.”

and

“Our measurements show that under positive bias, the radiative emission decay rate (Γ_{rad}) increases by 15% while under negative bias Γ_{rad} decreases by 11% (Fig. 5a). This amounts to the relative modulation of Γ_{rad} of 26% when applied gate voltage ranges between -1 V and $+1$ V (see Methods for further details). The measured voltage-dependent total emission decay rate (see Fig. 4c) and radiative decay rate can be used to determine variation of quantum yield of the QDs $\eta = \Gamma_{\text{rad}}/\Gamma_{\text{tot}}$ under an applied bias. We observe a 35% relative increase in quantum yield at an applied bias of $+1$ V and a 21% relative decrease in quantum yield at an applied bias of -1 V. This *in situ* control of quantum yield is a unique consequence of the bias-induced modulation of LDOS. Finally, we would like to emphasize that LDOS, radiative emission decay rate, and quantum yield do not depend on absorbance at the *excitation* wavelength of $\lambda=375$ nm.

To summarize, we observe that at a positive bias increase in PL intensity is always accompanied by an increase of both total and radiative emission decay rates, Γ_{tot} and Γ_{rad} (see Figs. 4 and 5). This implies that in addition to a reduced absorption of the ENZ-TiN/SiO₂/Ag plasmonic heterostructure at the QD

emission wavelength $\lambda=630$ nm and a slightly increased excitation intensity at $\lambda=375$ nm, LDOS modulation also contributes to an increase of PL intensity an under applied bias (even though, as it has been mentioned above, LODS modulation and absorbance modulation at QD emission wavelengths cannot be fully decoupled). This contrasts with the previously reported case of a bias-induced LDOS modulation, where an increase of LDOS has always been accompanied by a decrease in PL intensity, implying that primarily non-radiative emission decay rate has been modulated [8, 9].”

We previously clarified an approach used to calculate the quantum yield. The updated methodology to extract the radiative emission decay rate and quantum yield of QDs is described now in the Methods section.

“When calculating radiative emission decay rate, we take into account that our InP QDs are embedded in the SiO₂ layer of the ENZ-TiN/SiO₂/Ag heterostructure. As a result, a portion of the laser excitation ($\lambda=375$ nm) is going to be absorbed in the top TiN layer, which will affect the excitation intensity of the QDs. To estimate the effect of the possible variation of the excitation intensity, we measure the absorbance of the ENZ-TiN/SiO₂/Ag plasmonic heterostructure at an excitation wavelength of $\lambda=375$ nm under an applied bias (Figs. S13 and S14). As one can see, the absorbance stays almost constant for negative biases and shows a slight decrease for positive biases. Since absorption primarily occurs in the TiN layer, high absorbance results in the reduced excitation intensity of the QDs. Taking this into account, the bias dependent radiative emission decay rate (Γ_{rad}) can be given by the following formula [5, 10]:

$$\Gamma_{\text{rad}}(V)/\Gamma_{\text{rad}}^0 = (I_{\text{PL}}(V)/I_{\text{PL}}^0)[(1-A_{\text{laser}}(V))/(1-A_{\text{laser}}^0)]. \quad (1)$$

Here Γ_{rad}^0 is the radiative emission decay rate under zero bias, I_{PL}^0 is the peak PL intensity under zero bias, $I_{\text{PL}}(V)$ is the bias-dependent PL intensity, A_{laser}^0 is the absorbance in the ENZ-TiN/SiO₂/Ag heterostructure at an excitation wavelength of $\lambda=375$ nm at zero bias, and $A_{\text{laser}}(V)$ is the bias dependent absorbance at $\lambda=375$ nm. We would like to emphasize that absorbance in the TiN/SiO₂/Ag heterostructure primarily occurs in TiN layer (see Figs. S13 and S14). To calculate the bias-dependent quantum yield of our QDs embedded in TiN/SiO₂/Ag heterostructure, we use Eq. (1) and take into account that quantum yield of the emitter η is defined as a ratio of radiative and total decay rates $\eta=\Gamma_{\text{rad}}/\Gamma_{\text{tot}}$.”

By using the described approach, we have generated revised Fig. 5 and added error bars to all subfigures of Figure 5.

Figure 5 | Active control of quantum yield of QDs embedded in a gated plasmonic heterostructure. (a) Radiative decay rate (Γ_{rad}) of InP QDs (normalized to radiative decay rate at zero bias) embedded in the TiN/SiO₂/Ag plasmonic heterostructure as a function of gate voltage. Under positive bias Γ_{rad} shows a relative increase of 15% while under negative bias Γ_{rad} shows a relative decrease of 11%. (b) Dynamically tunable quantum yield of QDs (normalized to quantum yield at zero bias). When gate voltage V_G is increased from 0 V to +1 V, we observe a 35% relative increase of quantum yield. As gate voltage V_G varies from 0 V to -1 V, we observe a relative quantum yield decrease of 21%.

The error bars shown in Figs. 4 and 5 have been derived by using considerations described in the caption of Fig. R1.1.

Figure R1.1| PL intensity spectrum measured at the same excitation spot of the sample (curves 1, 2, 3, 4). The PL intensity measurement error is $\sim 2\%$. Since Figure 5 has been derived by calculating the ratio of PL spectra with and without an applied bias, the total measurement error bar in Figs. 4 and 5 is estimated to be $\sim 5\%$.

We added an additional figure to SI that that was used when calculating bias-dependent quantum yield of QDs.

Figure S14 | Lifetime at zero bias and absorbance at excitation and emission wavelengths. (a) By using time-resolved PL measurements, we identify the total decay of InP QDs at zero bias $\Gamma_{\text{tot}} = 2.56 \times 10^9$ s^{-1} . **(b)** Absorbance at the excitation wavelength ($\lambda = 375$ nm) in the ENZ-TiN/SiO₂/Ag heterostructure. **(c)** Absorbance at the emission wavelength ($\lambda = 630$ nm) in the ENZ-TiN/SiO₂/Ag heterostructure. Absorbance primarily occurs in the top TiN layer.

R1-2 The LDOS enhancement, as calculated and displayed in Fig. 4d is only valid for a z-oriented dipole. It is mentioned in the supplementary that a qualitatively similar result would be obtained in a more realistic case where dipoles are isotropically oriented. I believe it would be worth showing the corresponding LDOS enhancement for isotropic emitters as a function of wavelength, so that one will be able to compare this result to the values obtained experimentally in the paper. In other words, does the simulation indicate that we should effectively see a twofold enhancement in the radiative rate upon modulating the bias from -1V to +1V? Or is there any other competing effects at play?

We thank the reviewer for this comment. We replaced previously plotted z-projection of the $LDOS_z$ by an averaged out LDOS: $LDOS=(LDOS_x+ LDOS_y+ LDOS_z)/3$. The revised Fig. 4d plots averaged out LDOS enhancement.

The observed lifetime as well as the PL intensity and reflectance modulation reported in our work cannot be fully explained by changing carrier concentration in the Drude term of the Drude-Lorentz model that describes the dielectric permittivity of TiN. At high carrier concentrations ($N= 1.8 \times 10^{22} \text{ cm}^{-3}$), a number of different effects may contribute to the observed optical modulation, such as the dependence of the electron effective mass on the applied voltage due to the nonparabolicity of the conduction band and the dependence of the electron mobility on the applied bias. In Fig. 4d we assume that the dielectric permittivity of the 1 nm thick accumulation layer in the TiN is given by the measured dielectric permittivity of the TiN film with carrier concentration of $N= 4.1 \times 10^{22} \text{ cm}^{-3}$ (see Fig. 2). Strictly speaking, this assumption is not accurate; however, it allows us to estimate the sensitivity of the calculated LDOS with respect to the variation of the refractive index of the 1 nm thick accumulation layer (Fig. 4d). To summarize, we do not expect to have a quantitative match between our simulations and experimental data.

To address reviewer's concern we modify Fig. 4d.

Figure 4 | Gate-tunable spontaneous emission of QDs via modulation of the LDOS. (a). (b) (c) (d) Calculated LDOS enhancement spectra at the position of a QD (averaged over QD dipole orientations) for different carrier densities in a 1 nm thick modulated TiN layer. The black curve corresponds a homogeneous TiN film which is in the ENZ region ($N=1.8 \times 10^{22} \text{ cm}^{-3}$). The red curve corresponds to a TiN film with a 1 nm thick modulated TiN layer that is plasmonic but far from the ENZ region. The top panels show the simulated spatial distribution of the optical frequency electric field $|E|$ radiated by a QD ($\lambda=630$ nm). Both the calculated LDOS and optical field intensity $|E|$ in the SiO₂ gap increase with gate voltage.

We have also added the following paragraph to the main text of the manuscript:

“The observed lifetime as well as the PL intensity and reflectance modulation reported in our work cannot be fully explained by changing the carrier concentration in the Drude term of the Drude-Lorentz model that describes the dielectric permittivity of TiN. At high carrier concentrations ($N=1.8 \times 10^{22} \text{ cm}^{-3}$ for ENZ-TiN), a number of different effects may contribute to observed optical modulation, such as the dependence of the electron effective mass on the applied voltage due to the nonparabolicity of the conduction band and the dependence of the electron mobility on the applied bias. In Fig. 4d we assume that the dielectric permittivity of the 1 nm thick accumulation layer of TiN is given by the measured

dielectric permittivity of the TiN film with carrier concentration of $N = 4.1 \times 10^{22} \text{ cm}^{-3}$ (see Fig. 2). Strictly speaking, this assumption is not accurate; however, it allows us to estimate the sensitivity of the calculated LDOS with respect to variation of refractive index of the 1 nm thick accumulation layer (Fig. 4d).”

R1-3 The evolution of the imaginary part of permittivity against the carrier concentration, shown in Fig. 2 is rather strange. Indeed, the $\text{Im}(\epsilon)$ constantly increases with the concentration of free carriers but suddenly decreases between $N = 1.8 \times 10^{22}$ and $4.1 \times 10^{22} \text{ cm}^{-3}$. In addition, for this latter sample, the evolution of $\text{Im}(\epsilon)$ as a function of wavelength is not monotonic, what is also surprising if we compare to other studies. And finally, the fitting model for this sample is different: the authors used 3 Lorentz oscillators in that particular sample while they used 2 Lorentz oscillators for all other cases. This does not necessarily mean that the fit is wrong, but it is a strong indication that this particular sample is different than the others (probably due to structural differences between samples). Therefore, I strongly doubt a TiN thin film, initially prepared as ENZ-TiN, will effectively see its optical dispersion behaves similarly as in Fig. 2 upon injection of carriers through electrode gating. This is corroborated by results shown in Fig. S2: if the accumulation (depletion) of carriers in ENZ-TiN was qualitatively similar to changing the TiN from optically plasmonic to optically dielectric, the reflectance value of ENZ-TiN should oscillate between the respective ones of plasmonic and dielectric TiN (80% for optically plasmonic, 95% for optically dielectric, as displayed in Fig. S2). This is obviously not the case here, as the reflectance of ENZ-TiN under applied bias changes from ~67% to ~82%, what indicates that the effective changes in the optical dispersion of ENZ-TiN are different than the optical dispersions shown in Fig. 2. Therefore, I believe a better way of presenting things would be to show the optical modulation against the gate voltage as measured by ellipsometry (similarly to what was done by the authors in Nano letters 10 2111 (2010)).

Indeed, the imaginary part of ellipsometrically measured dielectric permittivity monotonically increases with the carrier concentration for carrier concentrations less than or equal to $1.8 \times 10^{22} \text{ cm}^{-3}$, and shows abrupt decrease for the sample with carrier concentration $4.1 \times 10^{22} \text{ cm}^{-3}$. However, we would like to point out that this kind of behavior has been previously reported in the literature (see Fig. 1b of Ref. [11]). In one case shown in Fig. 1b of Ref. [11], $\text{Im}(\epsilon)$ of “intermediately doped” TiN is higher than $\text{Im}(\epsilon)$ of “dielectric” TiN. On the other hand, over a broad wavelength range, $\text{Im}(\epsilon)$ of metallic TiN is smaller than $\text{Im}(\epsilon)$ of “intermediately doped” or “dielectric” TiN (see Fig 1b of Ref. [11]). In Ref. [11] it is argued that when oxygen impurity concentration in TiN is lowered, the crystallinity of TiN films is improved and

$R(\epsilon)$ becomes more negative and $\text{Im}(\epsilon)$ decreases. We agree with the reviewer's observation that when sputtering TiN films under different deposition conditions, the resulting TiN films will likely have compositional and structural differences. For example, the observed non-monotonic dependence of $\text{Im}(\epsilon)$ on the carrier concentration of the sputtered films gives an indirect evidence of compositional and structural differences between the films. Moreover, we performed compositional analysis of 135 nm thick TiN films deposited on Si substrate via DC sputtering. The sputtering conditions have been chosen to be identical to those used to deposit TiN in the tunable ENZ-TiN/SiO₂/Ag heterostructure. We found that the stoichiometry of our film is given as TiN_{0.8}O_{0.2}, where oxygen has been introduced into our film unintentionally. This result is consistent with previous reports [11, 12].

We thank the reviewer for asking about non-monotonic behavior of $\text{Im}(\epsilon)$ of TiN as a function of wavelength. We would like to address the question of how this compares with other studies. In the visible wavelength range, non-monotonic behavior of $\text{Im}(\epsilon)$ as a function of wavelength is commonly observed (see Fig. 1b of Ref. [11], Fig. 1c of Ref. [13], Fig. 1b of Ref. [14]). In our work we also observe non-monotonic behavior of $\text{Im}(\epsilon)$ as a function of wavelength only in the visible wavelength range. On the other hand, as we observe in our work, $\text{Im}(\epsilon)$ typically monotonically grows as a function of wavelength in the near-infrared wavelength range [11, 13, 14].

As we have mentioned when responding Question 2 of Reviewer 1, our calculations have shown that observed reflectance modulation cannot be fully explained by simply changing the carrier concentration in the Drude term of the Drude-Lorentz model that defines the dielectric permittivity of TiN. The carrier concentration of our ENZ-TiN film ($1.8 \times 10^{22} \text{ cm}^{-3}$) is at least 2 orders of magnitude greater than carrier concentrations that one typically deals with in ITO-based gate-tunable devices [15] ($3 \times 10^{20} \text{ cm}^{-3}$).

Hence, a number of different effects may contribute to the observed optical modulation such as dependence of electron effective mass on applied voltage due to nonparabolicity of conduction band and dependence of electron mobility on applied bias, etc. In Fig. S6 we assume that the dielectric permittivity of 1 nm thick accumulation layer in TiN is given by the measured dielectric permittivity of TiN film with carrier concentration of $N = 4.1 \times 10^{22} \text{ cm}^{-3}$ (see Fig. 2). As Reviewer 1 has pointed out, this assumption is, strictly speaking, not correct; however, it allows us to estimate the sensitivity of the calculated reflectance with respect to variation of the refractive index of 1 nm thick TiN layer which is located at the interface of TiN and SiO₂ (Fig. S6). To summarize, due to the high carrier concentration of TiN films, modifying the carrier concentration in the Drude term of the Drude-Lorentz model cannot reproduce experimentally observed reflectance modulation.

We would also like to address reviewer's comment regarding performing ellipsometry under applied bias. As Reviewer 1 has requested, we have performed ellipsometry of our ENZ-TiN/SiO₂/Ag heterostructure under applied bias. However, obtained data is not amenable to unambiguous interpretation due to reasons described below. The area of our gate-tunable ENZ-TiN/SiO₂/Ag heterostructure is only 1 mm². The reason why we choose such small device area is that the thickness of SiO₂ gate dielectric is only 9 nm. Larger area gate-tunable ENZ-TiN/SiO₂/Ag heterostructures would typically exhibit high leakage current upon biasing, due to defects in the gate dielectric. When performing ellipsometry it is important that incoming light is reflected only from the working area of the device. Hence, during ellipsometry measurements we had to use a focused light beam. However, the beam spot was still larger than the working area of the device, especially for larger incidence angles. By using this approach we have been able to obtain Ψ and Δ for different incidence angles. However, we haven't been able to obtain a reliable ellipsometry fit to extract the dielectric permittivity of TiN. Nevertheless, we have acquired Ψ and Δ for a given incidence angle (60°) and different applied biases. We observe optical modulation around 430 nm where Ψ and Δ exhibit maximum and minimum, respectively. Finally, we would like to note that since ellipsometry has been performed several months after fabricating the sample, the observed optical modulation is modest as compared to the result shown in Fig. 3 of the manuscript.

We combined the response to the current comment with the response to the Comment 3 of Reviewer 3, and added the following paragraph to our SI:

“When analyzing ellipsometry data, we used two or three Lorentz oscillators to fit the data obtained from the fabricated films. The key issue here is that optical properties of TiN films strongly depend on the film stoichiometry (N/Ti ratio), impurities (residual oxygen, post growth oxidation), grain size (which affects the mean free path of the conduction electrons) and density/porosity (which affect the conduction electron density) [13]. All these factors likely differ from one film to another resulting in different values of fitting parameters of a Drude-Lorentz model, such as electron mobility and frequencies of Lorentz oscillators. In particular, it has been previously shown that the spectral position of the zero crossing (ENZ point) is an indicator of film stoichiometry. It has been shown that for stoichiometric TiN films this crossing occurs at 2.65 eV (468 nm) [16]. As one can see from Fig. 2a, none of our films has an ENZ crossing at 468 nm. Hence, all films reported in this work are non-stoichiometric. Moreover, it has been recently shown that amount of residual oxygen in the sputtering chamber can dramatically affect optical properties of TiN films [11, 12, 14]. Based on these reports, we expect that each fabricated film has different chemical and structural composition. The review article [13] discusses the relation between the band structure of TiN obtained via density functional theory (DFT) calculations and the spectral positions of Lorentz oscillators in an ellipsometry fit. As one can see from Table 2 of the mentioned review article [13], the spectral

positions of Lorentz oscillators show significant variation. Moreover, the reported spectral position values do not always agree with DFT calculations. The reason for this is that the DFT calculations are performed for stoichiometric TiN while the films shown in our work as well as a number of films shown in the review article [13] are non-stoichiometric. Moreover, we performed compositional analyses on 135 nm-thick TiN films deposited on Si substrates via DC sputtering. The sputtering conditions were chosen to be identical to those used to deposit TiN in the tunable ENZ-TiN/SiO₂/Ag heterostructure. We found that the stoichiometry of our film is given as TiN_{0.8}O_{0.2}, where oxygen has been introduced into our film unintentionally. This result is consistent with previous reports [11, 12]. Hence, most likely, 7 nm-thick TiN films incorporated in our TiN/SiO₂/Ag films also include a significant amount of oxygen impurities. Based on this, it is not straightforward to relate positions of Lorentz oscillators to the band structure of our TiN, due to limited knowledge of the band structure of the actual film. To summarize, the choice of the model that we use to fit the ellipsometry data (two vs. three Lorentz oscillators, etc.) is largely dictated by the composition and structure of the films, which varies. Due to a limited knowledge of the dependence of the TiN band structure on the material stoichiometry and the amount of incorporated oxygen impurities, we are not able to perform a direct comparison between the positions of Lorentz oscillators and the material band structure.

One can obtain indirect evidence of compositional and structural differences in sputtered TiN films when analyzing the dependence of the imaginary part of the measured dielectric permittivity $\text{Im}(\epsilon)$ on the carrier concentration of the films. Indeed, $\text{Im}(\epsilon)$ monotonically increases with the carrier concentration for carrier concentrations less than or equal to $1.8 \times 10^{22} \text{ cm}^{-3}$ and abruptly decreases for the sample with carrier concentration $1.8 \times 10^{22} \text{ cm}^{-3}$. This kind of behavior has been previously reported in literature [11]. In the mentioned paper, $\text{Im}(\epsilon)$ of “intermediately doped” TiN is greater than $\text{Im}(\epsilon)$ of “dielectric” TiN [11]. On the other hand, over a broad wavelength range, $\text{Im}(\epsilon)$ of “metallic” TiN is smaller than $\text{Im}(\epsilon)$ of “intermediately doped” or “dielectric” TiN [11]. It is argued that when the oxygen impurity concentration in TiN is lessened, the crystallinity of TiN films is improved, $\text{Re}(\epsilon)$ becomes more negative, and $\text{Im}(\epsilon)$ decreases.

Interestingly, in the visible wavelength range we observe a non-monotonic behavior of $\text{Im}(\epsilon)$ as a function of wavelength (Fig. 2b) that is consistent with previously reported functional dependences of $\text{Im}(\epsilon)$ on wavelength [11, 13, 14]. On the other hand, as we observe in our work, $\text{Im}(\epsilon)$ typically grows monotonically as a function of wavelength in the near-infrared wavelength range [11, 13, 14].”

We added the following paragraph and Figure S2 into SI:

“We have also performed ellipsometry of our ENZ-TiN/SiO₂/Ag heterostructure under applied bias. However, obtained data is not amenable to unambiguous interpretation due to reasons described below. The area of our gate-tunable ENZ-TiN/SiO₂/Ag heterostructure is only 1 mm². The reason why we choose such small device area is that the thickness of SiO₂ gate dielectric used in our study is only 9 nm. Larger area gate-tunable ENZ-TiN/SiO₂/Ag heterostructures would typically exhibit high leakage current upon biasing due to defects of gate dielectric. When performing ellipsometry it is important that incoming light is reflected from the working area of the device only. Hence, during ellipsometry measurements we had to use focused light beam. However, the beam spot was still larger than the working area of the device, especially for larger incidence angles. By using this approach we have been able to obtain Ψ and Δ for different incidence angles. However, we haven’t been able to obtain a reliable ellipsometry fit to extract dielectric permittivity of TiN. Nevertheless, we have acquired Ψ and Δ for a given incidence angle (60°) and different applied biases. We observe optical modulation around 430 nm where Ψ and Δ exhibit maximum and minimum, respectively. Finally, we would like to note that since ellipsometry has been performed several months after fabricating the sample, the observed optical modulation is modest as compared to the result shown in Fig. 3 of the manuscript.”

Figure S2 | Reflectance modulation measured via focused beam spectroscopic ellipsometry. Δ and Ψ for the ENZ-TiN/SiO₂/Ag heterostructure as a function of wavelength at different voltages. In this measurement, the incidence angle is 60°. We observe modulation of Ψ and Δ under applied electrical bias. When electrical bias increases from -1 V to +1 V, we observe a monotonic increase of Ψ and monotonic decrease of Δ . Inset shows the SEM image of the device and the normal incident beam size of focused beam spectroscopic ellipsometry.

We also added the following paragraph to SI:

“Our calculations show that observed reflectance modulation cannot be fully explained by simply changing the carrier concentration in the Drude term of the Drude-Lorentz model that defines the dielectric permittivity of TiN. The carrier concentration of our ENZ-TiN film ($1.8 \times 10^{22} \text{ cm}^{-3}$) is at least 2 orders of magnitude higher as compared to carrier concentrations that one typically deals with in ITO-based gate-tunable devices [15] ($3 \times 10^{20} \text{ cm}^{-3}$). Therefore, a number of different effects may contribute to the observed optical modulation. One possible effect is the electron effective mass’s dependence on applied voltage due to nonparabolicity of conduction band. Another possible contributor is the dependence of the electron mobility on applied bias. In Fig. S6 we have assumed that the dielectric permittivity of 1 nm thick accumulation layer in TiN is given by the measured dielectric permittivity of TiN film with carrier concentration of $N = 4.1 \times 10^{22} \text{ cm}^{-3}$ (see Fig. 2). This assumption is, strictly speaking, not correct; however, it allows us to estimate the sensitivity of the calculated reflectance with respect to variation of refractive index of 1 nm thick TiN layer which is located at the interface of TiN and SiO₂ (Fig. S6). To summarize, due to high carrier concentration of TiN films, modifying carrier concentration in the Drude term of the Drude-Lorentz model cannot reproduce experimentally observed reflectance modulation.”

R1-4 There are 4 TiN samples displayed in Fig. 2 and the corresponding material properties values of all 4 are shown in table S1, but in the Drude-Lorentz fitting parameters table S2, only 3 of them are shown. Why? This is especially surprising given that the missing sample is the most important one, i.e. the ENZ-TiN that is used for all of the modulation devices. This data needs to be included in the supplementary.

We revised table S2 and added a line with Drude-Lorentz parameters that we used to define dielectric permittivity of ENZ-TiN in ENZ-TiN/SiO₂/Ag heterostructure.

As we have mentioned in our response to Question 3 of Reviewer 1, because of small device area, we are not able to obtain reliable ellipsometry fits for ENZ-TiN which is incorporated in ENZ-TiN/SiO₂/Ag heterostructure. We have identified dielectric permittivity of TiN in TiN/SiO₂/Ag heterostructure by fitting parameters of Drude-Lorentz model to reflectance spectrum. The reflectance spectrum was taken under normal incidence with 5X objective (Olympus, NA=0.14) focusing down a supercontinuum laser (Fianium) to a small spot of 3 μm in diameter, which is much smaller than the working area of our device (1 mm^2). When fitting parameters of Drude-Lorentz model to reflectance data, we have used electron mobility and carrier concentration values derived via Hall measurements (see Table S1). As one can see from Fig. S3, the calculated and measured reflectance are in good agreement.

We revised Fig. 2 by incorporating into it newly derived dielectric permittivity for ENZ-TiN. As one can see from Fig. 2, real part of dielectric permittivity of our TiN shows double-ENZ crossing that is in good agreement with recently reported results [14].

We revised Table S2:

“Table S2 | The Drude-Lorentz parameters for the complex dielectric permittivity of TiN. These values have been used to produce Fig. 2.

N (cm ⁻³)	E_p (eV)	Γ (eV)	f_1	E_{o1} (eV)	γ_1 (eV)	f_2	E_{o2} (eV)	γ_2 (eV)	f_3	E_{o3} (eV)	γ_3 (eV)	ϵ_∞
5.9×10^{20}	–	–	5.9	3.8	2.1	3.8	1.2	1.6	–	–	–	2.0
2.6×10^{21}	4.6	17.3	13.3	1.0	1.8	6.6	4.0	2.0	–	–	–	1.7
1.8×10^{22}	5.15	102	22	0.6	1.6	10	4.0	0.8	–	–	–	4
4.1×10^{22}	6.9	1.04	0.4	2.4	0.3	8.9	3.5	2.4	1.1	2.6	0.7	1.3

“

We added to SI the following paragraph:

“As we have previously mentioned, because of small device area, we are not able to obtain reliable ellipsometry fits for ENZ-TiN which is incorporated in ENZ-TiN/SiO₂/Ag heterostructure. We have identified dielectric permittivity of TiN in TiN/SiO₂/Ag heterostructure by fitting parameters of Drude-Lorentz model to reflectance spectrum. The reflectance spectrum was taken under normal incidence with 5X objective (Olympus, NA=0.14) focusing down a supercontinuum laser (Fianium) to a small spot of 3 μ m in diameter, which is much smaller than the working area of our device (1 mm²). When fitting parameters of Drude-Lorentz model to reflectance data, we have used electron mobility and carrier concentration values derived via Hall measurements (see Table S1). As one can see from Fig. S3, the calculated and measured reflectance are in good agreement. We revised Fig. 2 by incorporating into it newly derived dielectric permittivity for ENZ-TiN. As one can see from Fig. 2, real part of dielectric permittivity of our TiN shows double-ENZ crossing that is in good agreement with recently reported results [14].”

Figure S3 | Measured and calculated reflectance of ENZ-TiN/SiO₂/Ag heterostructure. The dielectric permittivity of ENZ-TiN is extracted from reflectance measurements.

We have also modified Figure 2:

Figure 2 | Optical properties of TiN films. Measured (a) real and (b) imaginary parts of the complex dielectric permittivity of TiN thin films. The fabricated TiN films are *n*-type with carrier densities ranging from 5.9×10^{20} to $4.1 \times 10^{22} \text{ cm}^{-3}$. Depending on the carrier density, the fabricated TiN films can be optically dielectric ($\text{Re}(\epsilon) > 0$) or optically plasmonic ($\text{Re}(\epsilon) < 0$). The gray dotted line in (a) denotes $\text{Re}(\epsilon) = 0$. For a carrier density of $1.8 \times 10^{22} \text{ cm}^{-3}$, $\text{Re}(\epsilon)$ is in the ϵ -near-zero (ENZ) region ($-1 < \text{Re}(\epsilon) < 1$) over the entire wavelength range. Hence, by applying a voltage between TiN and Ag (Fig. 1), the interfacial TiN film region undergoes a transition from an optically dielectric to an optically plasmonic state, or vice versa. For comparison, we also plot the dielectric permittivity values for gold and silver (Johnson & Christy) [17] two standard plasmonic materials, which are shown in dashed lines.

R1-5 The dynamic modulation is convincingly shown in Fig. S3 but there are no indications on the modulation amplitude. It should be explicitly mentioned in the paper. Also, why is the applied bias $\pm 5V$ in that case while it is $\pm 1V$ throughout the manuscript? What happens in the device containing QDs when the voltage is increased to voltages higher than 1 V. The authors should explain that in the manuscript as it could be a limitation to potential applications.

We thank the reviewer for pointing this out. We performed modulation frequency measurements four months after the sample was fabricated. Therefore, our sample had already degraded. Figure S5 shows the reflectance spectrum of the degraded sample taken right before modulation frequency measurements. Degraded samples typically exhibit an increased contact resistance between the TiN and contact electrode. That's why, we believe, we need to apply a larger electrical bias of ± 5 V to observe reflectance modulation. We would like to point out that the application of higher voltages does not necessarily imply that the built-in electric field inside the heterostructure is going to be larger. Moreover, reflectance characteristics of the degraded sample have also been modified; likely, due to sample oxidation (compare Figs. 3a and S5). Despite sample degradation, the sample still exhibits optical modulation under applied bias. By using the degraded sample we performed reflectance modulation measurements. Importantly, we observe modulation frequency of 20 MHz, and the detected reflectance values show perfect match with the reflectance values measured in Fig. S5. Finally, we would like to note that the samples can be protected by depositing a thin passivation layer (e.g., Al_2O_3).

Finally, we would like to point out that within the scope of our project we avoided applying voltages with absolute values larger than 1 V because we were worried to burn our samples.

Following reviewer's request, we marked the measured reflectance values at y-axis of Fig. S5 in the revised SI.

Figure S5 | Modulation speed of TiN/SiO₂/Ag heterostructure. Reflectance of TiN/SiO₂/Ag heterostructure under time-varying bias that changes between -5 V and 5 V with a frequency of 20 MHz. The modulation amplitude varies between 50% and 63%. Our detection frequency was limited by the Si photodetector response time. This implies that modulation frequency of our device could potentially be higher than 20 MHz.

In addition, we've added a description on sample degradation in supplementary section 4:

“We performed modulation frequency measurements four months after the sample was fabricated. As a result, at the time of modulation frequency measurements, our sample was already degraded. Figure S5 shows reflectance spectrum of the degraded sample taken right before modulation frequency measurements. Degraded samples typically exhibit an increased contact resistance between the TiN and contact electrode. That's why, we believe, we need to apply a larger electrical bias of ± 5 V to observe reflectance modulation. We would like to point out that the application of higher voltages does not necessary imply that the built-in electric field inside the heterostructure is going to be larger. Moreover, the reflectance characteristics of the degraded sample have also been modified; likely, due to sample oxidation (compare Figs. 3a and S5). Despite sample degradation, the sample still exhibits optical modulation under applied bias. We performed reflectance modulation measurements using the degraded sample. Importantly, we observe a modulation frequency of 20 MHz, and the detected reflectance values show a perfect match with the reflectance values measured in Fig. S5. Finally, we would like to note that the samples can be protected by depositing a thin encapsulation layer (e.g., Al₂O₃).”

Detailed Response to Reviewer #2's Comments:

In this manuscript, the authors experimentally demonstrated dynamic control of the visible spontaneous emission of colloidal quantum dots by electrically tuning the local optical environment. Specifically, by carefully changing the doping level of TiN, its permittivity can reach near zero (ENZ) at the visible wavelength regime. Using external voltage to build up a charge depletion or accumulation layer in TiN, its refractive index will be greatly tuned near ENZ wavelength, thus leading to the LDOS change of QDs and resulting in the modulation of spontaneous emission.

This manuscript is well written and the results are technologically sound. Electrically active control of the visible light emission have great potential applications. The layered structure in this work can be easily fabricated and the dynamic modulation through LDOS could reach ultrafast speed. I believe it is a good contribution to the field of optoelectronics. Thus, I recommend it to be considered for publication in Nature Communications after addressing the following concerns.

R2-1 While discussing the photoluminescence (PL) intensity modulation (i.e., lines 88-99 of page 4), the authors attributed the modulation to LDOS change caused by varied external voltages (Fig. 4a). However, although the external optical pump power was constant for varied voltages, the actual pump field intensity at the emitter locations might still vary for different voltages, thus leading to modulation of PL intensity as well. Although the lifetime results (based on LDOS) in Fig. 4c can partially explain the modulated PL intensity, the pump effect may also contribute to such modulation. As shown in Fig. 3a, the reflectance varies (in wide spectral range ~450 to 800nm) for different external voltages. The authors should comment on this pump effect of 375nm excitation wavelength in the main text.

We thank the reviewer for this valuable comment. Indeed, a possible dependence of the excitation field intensity on applied bias needs to be taken into account, when analyzing the reasons behind observed PL intensity modulation. We measured the absorbance of ENZ-TiN/SiO₂/Ag plasmonic heterostructure as a function of applied bias at the laser excitation wavelength of $\lambda=375$ nm. Figure S13 shows a slight decrease in absorbance at positive voltages that implies increased excitation intensity. This slight change in absorbance will definitely contribute to the observed enhancement of PL intensity for positive biases. Note, however, that LDOS enhancement, radiative emission decay rate, and quantum yield do not depend on absorbance modulation at the *excitation* wavelength of $\lambda=375$ nm. In our manuscript, we have shown

that the radiative emission decay rate and quantum yield increase at positive biases (Fig. 5). Hence, LDOS modulation plays an important role in the observed modulation of PL intensity.

We combined the response to the current question with the response to Question 1 of Reviewer 1, and added the following discussion to the main text of the manuscript:

“Thus, in our experiment we observe a simultaneous increase (decrease) of PL intensity and total decay rate of emission Γ_{tot} . However, it still needs to be proven that the measured LDOS modulation contributes to the observed PL intensity modulation. For example, variation of excitation field intensity under applied bias could also result in modulation of PL intensity. To investigate this possibility, we measure the absorbance spectrum of an ENZ-TiN/SiO₂/Ag plasmonic heterostructure at a laser excitation wavelength of $\lambda=375$ nm (Fig. S13a) as a function of applied voltage. We observe a slight decrease of absorbance at positive voltages that implies increased excitation intensity, and consequently, increased PL intensity. Observed PL intensity modulation can also be attributed to reduction or increase of absorbance of ENZ-TiN/SiO₂/Ag plasmonic heterostructure at QD emission wavelengths (Fig. S13b). However, we would like to point out that absorbance modulation at QD emission wavelength and modulation of the total decay rate of a QD are interrelated since QDs are placed in the immediate vicinity of the TiN layer (see Fig. 1). In what follows, we further investigate how LDOS modulation contributes to the observed PL modulation.”

and

“Our measurements show that under positive bias the radiative emission decay rate (Γ_{rad}) increases by 15% while under negative bias Γ_{rad} decreases by 11% (Fig. 5a). This amounts to a relative modulation of Γ_{rad} of 26% when the applied gate voltage varies between -1 V and $+1$ V (see Methods for further details). The measured voltage-dependent total emission decay rate (see Fig. 4c) and radiative decay rate can be used to determine the variation of QD quantum yields $\eta=\Gamma_{\text{rad}}/\Gamma_{\text{tot}}$ under applied bias. We observe a 35% relative increase of quantum yield at an applied bias of $+1$ V and 21% relative decrease of quantum yield at an applied bias of -1 V. This *in situ* control of quantum yield is a unique consequence of the bias-induced modulation of LDOS. We emphasize that LDOS, radiative emission decay rate and quantum yield do not depend on the absorbance at the excitation wavelength of $\lambda=375$ nm.

To summarize, we observe that at positive bias the increase in PL intensity is always accompanied by increases in both total and radiative emission decay rates: Γ_{tot} and Γ_{rad} (see Figs. 4 and 5). This implies

that besides reduced absorption of the ENZ-TiN/SiO₂/Ag plasmonic heterostructure at the QD emission wavelength $\lambda=630$ nm and slightly increased excitation intensity at $\lambda=375$ nm, LDOS modulation also contributes to an increase in PL intensity under applied bias (even though, as it has been mentioned above, LDOS modulation and absorbance modulation at QD emission wavelengths cannot be fully decoupled). This contrasts to previously reported cases of bias-induced LDOS modulation where an increase of LDOS has always been accompanied by a decrease in PL intensity which implies that the applied bias primarily affected the non-radiative decay rate [8, 9].”

Figure S13 | Absorbance of ENZ-TiN/SiO₂/Ag plasmonic heterostructure. (a) Measured absorbance as a function of applied bias at the laser excitation wavelength of $\lambda=375$ nm. (b) Measured absorbance spectrum of ENZ-TiN/SiO₂/Ag plasmonic heterostructure for different applied biases. Our calculations show that absorption primarily occurs in top TiN layer.

R2-2 In the section “Active control of quantum yield of QDs” (lines 109-123), the authors only briefly summarized the results of radiative/non-radiative rate and quantum yields. A little more discussion about the observed results may be helpful for the readers. In addition, although the experimental details for this section are covered in the Supplementary Materials, it may be better to add some descriptions in either the main text or the Methods section.

We agree with this comment of the reviewer. In the revised version of the manuscript, we expanded the discussion in the section “Active control of quantum yield of QDs”. We also clarified our calculation method of quantum yield of QDs. We added the updated calculation method to the Methods section.

We added the following discussion in the section “Active control of quantum yield of QDs” which has also been cited when answering Question 1 of Reviewer 2:

“Our measurements show that under positive bias the radiative emission decay rate (Γ_{rad}) increases by 15% while under negative bias Γ_{rad} decreases by 11% (Fig. 5a). This amounts to a relative modulation of Γ_{rad} of 26% when the applied gate voltage varies between -1 V and $+1$ V (see Methods for further details). The measured voltage-dependent total emission decay rate (see Fig. 4c) and radiative decay rate can be used to determine the variation of QD quantum yields $\eta = \Gamma_{\text{rad}}/\Gamma_{\text{tot}}$ under applied bias. We observe a 35% relative increase of quantum yield at an applied bias of $+1$ V and 21% relative decrease of quantum yield at an applied bias of -1 V. This *in situ* control of quantum yield is a unique consequence of the bias-induced modulation of LDOS. We emphasize that LDOS, radiative emission decay rate and quantum yield do not depend on the absorbance at the excitation wavelength of $\lambda = 375$ nm.

To summarize, we observe that at positive bias the increase in PL intensity is always accompanied by increases in both total and radiative emission decay rates: Γ_{tot} and Γ_{rad} (see Figs. 4 and 5). This implies that besides reduced absorption of the ENZ-TiN/SiO₂/Ag plasmonic heterostructure at the QD emission wavelength $\lambda = 630$ nm and slightly increased excitation intensity at $\lambda = 375$ nm, LDOS modulation also contributes to an increase in PL intensity under applied bias (even though, as it has been mentioned above, LDOS modulation and absorbance modulation at QD emission wavelengths cannot be fully decoupled). This contrasts to previously reported cases of bias-induced LDOS modulation where an increase of LDOS has always been accompanied by a decrease in PL intensity which implies that the applied bias primarily affected the non-radiative decay rate [8, 9].”

We revised our methodology to calculate the radiative emission decay rate and the quantum yield of quantum dots and added the description of our approach to the Methods section:

“When calculating the radiative emission decay rate we take into account that our InP QDs are embedded in the SiO₂ layer of the ENZ-TiN/SiO₂/Ag heterostructure. As a result, a portion of the laser excitation ($\lambda = 375$ nm) is going to be absorbed in the top TiN layer and will affect the excitation intensity of the QDs. To estimate the effect of variation of excitation intensity we measure the absorbance of ENZ-TiN/SiO₂/Ag plasmonic heterostructure at an excitation wavelength of $\lambda = 375$ nm under an applied bias (Figs. S13 and S14). As one can see, absorbance stays almost constant for negative biases and shows a slight decrease for positive biases. Since absorption primarily occurs in the TiN layer, high absorbance results in a reduced excitation intensity of the QDs. Taking this into account, one can write the bias dependent radiative emission decay rate (Γ_{rad}) as [10, 11]:

$$\Gamma_{\text{rad}}(V)/\Gamma_{\text{rad}}^0 = (I_{\text{PL}}(V)/I_{\text{PL}}^0)[(1-A_{\text{laser}}(V))/(1-A_{\text{laser}}^0)], \quad (1)$$

where Γ_{rad}^0 is the radiative emission decay rate under zero bias, I_{PL}^0 is the peak PL intensity under zero bias, $I_{\text{PL}}(V)$ is the bias-dependent PL intensity, A_{laser}^0 is the absorbance in the ENZ-TiN/SiO₂/Ag heterostructure at an excitation wavelength of $\lambda=375$ nm at zero bias, and $A_{\text{laser}}(V)$ is the bias dependent absorbance at $\lambda=375$ nm. We would like to emphasize that absorption in the TiN/SiO₂/Ag heterostructure primarily occurs in the TiN layer (see Figs. S13 and S14). To calculate the bias-dependent quantum yield of our QDs embedded in the TiN/SiO₂/Ag heterostructure, we use Eq. (1) and take into account that the quantum yield of the emitter, η , is defined as the ratio of radiative and total decay rates $\eta=\Gamma_{\text{rad}}/\Gamma_{\text{tot}}$.

Detailed Response to Reviewer #3's Comments:

The manuscript presented by Lu et al reports the demonstration of an interesting and long-sought-after mechanism to control the spontaneous emission yield of InP/ZnS core-shell nanostructures by dynamically controlling the local density of photonic modes they experience using an electrically tunable nano-plasmonic device. The use of the TiN layer as a tunable plasmonic material, having a plasma frequency in the visible range, is an interesting result that will certainly be of interest to the community of researchers working in photonics and plasmonics. However, the observed effect is weak, consisting of a $\pm 7\%$ change in the radiative luminescence efficiency as the voltage applied to the field-effect capacitor is tuned from -1V to +1V. Moreover, whilst an impressive array of supplementary material is presented that certainly helps to support the conclusions that the spontaneous emission rate, and hence the quantum efficiency, of the colloidal QDs is indeed varied by tuning the electric field, but I have some questions for the authors that should be addressed before publication is further considered. I feel that the topic of the paper is of sufficient interest and timeliness to warrant publication in nature communications, providing that the technical concerns raised below are fully addressed.

R3-1 The predictions of fig 4d would indicate that one might expect a tunable enhancement of the LDOS over the spectral range between ca 550-750nm, with a maximum response occurring around ~600nm, but active over the whole of this spectral range. In the PL spectra presented in the inset of fig 4a, the field induced emission enhancement seems to be only present in a much narrower spectral range 600-660nm. How do the authors account for this discrepancy?

We appreciate the reviewer's comment. Indeed, our simulations show fairly broadband tunable LDOS enhancement. However, in the experiment, we observe tunable PL intensity in a much narrower spectral range centered around 630 nm. The reasons behind this apparent discrepancy can be understood by recalling that in our experiment we measure a QD ensemble PL intensity spectrum that is defined by i) the size distribution of QDs and ii) intrinsic single emitter emission spectrum. In fact, the observed wide breadth of the QD ensemble emission spectrum is due to different sizes of the involved QDs (inhomogeneous broadening). The size distribution of QDs in the purchased QD ensemble is such that a large fraction of QDs emits at 630 nm. If we assume that PL intensity modulation of each individual QD does not depend on QD emission wavelength, one may expect that the largest modulation of the ensemble PL spectrum will occur around the wavelength of $\lambda=630$ nm since the number of individual QDs emitting around this particular wavelength is very high.

We added the following clarifying paragraph to the main text of the manuscript:

“The measured PL intensity spectrum shows significant modulation only around the central emission wavelength of $\lambda=630$ nm. This apparently contradicts the theoretical prediction of the broadband LDOS modulation under applied bias (Fig. 4d). The contradiction is resolved by recalling that the measured PL intensity spectrum originates from an inhomogeneously broadened QD ensemble. Note that the relative PL intensity modulation of each QD does not depend on the emission wavelength (see Fig. S7c). In the measured ensemble, the size distribution of the QDs is such that a large fraction of the individual QDs emits around the wavelength of $\lambda=630$ nm yielding a brighter PL signal at $\lambda=630$ nm.”

R3-2 There does seem to be a shift of the peak position of the PL data presented in the inset of fig 4a upon modifying the applied voltage. Large static electric fields can impact on the average charge status of the quantum dots and such effects can give rise to both spectral shifts and a change in the radiative efficiency. How can the authors discount the possibility that the charge status of the dots are tuned by the electric field, giving rise to the observed change in radiative efficiency? Here, perhaps it would help to present the raw PL data more prominently (larger panel, logarithmic scale, differential spectra recorded with a gate voltage V , relative to the spectra obtained at $V=0$...). This is to my mind a crucial point, since the spectral dependence of the LDOS modulation / enhancement for ENZ TiN and Optically Plasmonic TiN is likely to be of significant interest to readers and the result should be unambiguous.

We appreciate the reviewer's question regarding the possible impact of large static electric fields on emission properties of QDs. We believe that the performed control experiments eliminate the possibility that experimentally observed modulation of emission properties of QDs is due to static field-induced modification of the charge status of QDs.

- a) We observe ***no spectral shift*** of the peak position in the presented PL data when electrical bias is applied between Ag and TiN. Moreover, we observe no spectral shift in the PL data both in the case of the ENZ-TiN/SiO₂/Ag sample and the Ti/SiO₂/Ag control sample (Figs. S7 and S8). To make this more evident, we plot the position of the PL peak as a function of applied bias for the ENZ-TiN/SiO₂/Ag sample and the Ti/SiO₂/Ag (Figs. S7 and S8).
- b) We observe ***no modulation of PL intensity spectrum*** for quantum dots embedded in Ti/SiO₂/Ag heterostructure when Ti and Ag are biased with respect to one another (Fig. S8).
- c) We observe ***no lifetime modulation*** of QDs embedded in Ti/SiO₂/Ag heterostructure when Ti and Ag are biased with respect to one another.

Points a)-c) prove that emission properties of InP/ZnS QDs are unaffected by applied static electric fields of the order of 1.1 MV/cm. In other words, our InP/ZnS QDs show no quenching or red-shift of emission, which is characteristic for cadmium-based core-shell colloidal QDs [18, 19]. This is attributable to the large bandgap difference between InP core and ZnS shell materials.

Finally, we note that we observe an increase or decrease of PL intensity depending on the polarity of the electric field. If the observed modulation of PL intensity were caused by the change of the internal state of QDs, the PL intensity would only depend on the absolute value of the electric field and not on its direction. Moreover, when the internal state of a QD is modified due to static electric field, PL quenching is observed [18, 19]. In our case, we observe an increase of PL intensity for positive biases.

Thus, the observed modulation of radiative emission decay rate Γ_{rad} is not due to the change of the charge status of QDs.

Figure S7 | Modulation of PL intensity of InP QDs embedded in the gated TiN/SiO₂/Ag active plasmonic heterostructure. (a) PL intensity spectra for different gate voltages in a linear scale (the same PL spectra we show in Fig. 4a). (b) PL spectra in a log scale. (c) Differential PL spectra. Red line: $(\text{PL}(1\text{V}) - \text{PL}(0\text{V}))/\text{PL}(0\text{V})$ and blue line: $(\text{PL}(-1\text{V}) - \text{PL}(0\text{V}))/\text{PL}(0\text{V})$. (d) PL peak position as a function of applied voltage. We observe no detectable wavelength shift.

Figure S8 | Modulation of the PL intensity of InP QDs embedded in the gated Ti/SiO₂/Ag passive heterostructure. (a) PL intensity spectra for different gate voltages in a linear scale (the same PL spectra we show in Fig. 4b). (b) PL spectra in a log scale. (c) Differential PL spectra. Red line: $(PL(1V) - PL(0V))/PL(0V)$ and blue line: $(PL(-1V) - PL(0V))/PL(0V)$. (d) PL peak position as a function of applied voltage. We observe no detectable wavelength shift.

We added the following clarifying sentences to the manuscript:

“In our experiments we observe no shift of the wavelength of PL peak intensity under applied bias in both cases when QDs are embedded in a tunable TiN/SiO₂/Ag or passive Ti/SiO₂/Ag heterostructure (see Figs. S7 and S8).”

and

“The fact that we observe both an increase and decrease of PL intensity, depending on the polarity of the electric field, provides additional evidence that the observed modulation of PL signal is not caused by the change of the internal state of the QDs under applied bias. If this were the case, the observed modulation of PL intensity would depend only on the magnitude of the electric field and not on its direction.”

Figure S7 and Figure S8 have been added to SI.

R3-3 The use of either two or three Lorentz oscillators to fit the ellipsometry data and produce the carrier density dependent dielectric function (fig 2) seems to be somewhat arbitrary? Could the authors please relate the frequencies of the Lorentz oscillators to the expected bandstructure of the n-doped TiN and explain the rationale behind the choice of using either 2 or 3 Lorentz oscillators to fit the dielectric function as the free carrier density N varies?

We thank the reviewer for her or his comment. When analyzing ellipsometry data, we use two or three Lorentz oscillators to fit the data obtained from the fabricated films. The key issue here is that optical properties of TiN films strongly depend on the film stoichiometry (N/Ti ratio), impurities (residual oxygen, post growth oxidation), grain size (which affects the mean free path of the conduction electrons) and density/porosity (which affect the conduction electron density) [13]. All these factors likely differ from one film to another resulting in different values of fitting parameters of Drude-Lorentz model, such as electron mobility and frequencies of Lorentz oscillators. In particular, it has been previously shown that the spectral position of the zero crossing (ENZ point) is an indicator of film stoichiometry. It has been shown that for stoichiometric TiN films this crossing occurs at 2.65 eV (468 nm) [16]. As one can see from Fig. 2a, none of our films has an ENZ crossing at 468 nm. Hence, all films reported in this work are non-stoichiometric. Moreover, it has been recently shown that amount of residual oxygen in the sputtering chamber can dramatically affect optical properties of TiN films [11, 12, 14]. Based on these reports, we expect that each fabricated film has different chemical and structural composition. The review article [13] discusses the relation between the band structure of TiN obtained via density functional theory (DFT) calculations and the spectral positions of Lorentz oscillators in an ellipsometry fit. As one can see from Table 2 of the mentioned review article [13], the spectral positions of Lorentz oscillators show significant variation. Moreover, reported spectral position values do not always agree with DFT calculations. The reason for this is that the DFT calculations are performed for stoichiometric TiN while the films shown in our work as well as a number of films shown in the review article [13] are non-stoichiometric. Moreover, we performed compositional analysis on a 135 nm-thick TiN film deposited on Si substrates via DC sputtering. The sputtering conditions were chosen to be identical to those used to

deposit TiN in the tunable ENZ-TiN/SiO₂/Ag heterostructure. We found that the stoichiometry of our film is given as TiN_{0.8}O_{0.2}, where oxygen has been introduced into our film unintentionally. This result is consistent with previous reports [11, 12]. Hence, most likely, the 7 nm-thick TiN films incorporated in our TiN/SiO₂/Ag films also include a significant amount of oxygen impurities. Based on this, it is not straightforward to relate the position of Lorentz oscillators to the band structure of TiN, due to limited knowledge of the band structure of the actual film. To summarize, the choice of the model that we use to fit the ellipsometry data (two vs. three Lorentz oscillators, etc.) is largely dictated by compositional and structural differences between films. Due to a limited knowledge of the dependence of the TiN band structure on the material stoichiometry and the amount of incorporated oxygen impurities, we are not able to perform direct comparison between the positions of Lorentz oscillators and the material band structure.

One can obtain indirect evidence of compositional and structural differences between sputtered TiN films when analyzing the dependence of the imaginary part of the measured dielectric permittivity $\text{Im}(\epsilon)$ on the carrier concentration of the films. Indeed, $\text{Im}(\epsilon)$ monotonically increases with the carrier concentration for carrier concentrations smaller or equal to $1.8 \times 10^{22} \text{ cm}^{-3}$ and abruptly decreases for the sample with carrier concentration $1.8 \times 10^{22} \text{ cm}^{-3}$. This kind of behavior has been previously reported in the literature [11]. In the mentioned paper, $\text{Im}(\epsilon)$ of “intermediately doped” TiN is greater than $\text{Im}(\epsilon)$ of “dielectric” TiN [11]. On the other hand, over a broad wavelength range, $\text{Im}(\epsilon)$ of “metallic” TiN is smaller than $\text{Im}(\epsilon)$ of “intermediately doped” or “dielectric” TiN [11]. It is argued that when the oxygen impurity concentration in TiN is lessened, the crystallinity of TiN films is improved, $R(\epsilon)$ becomes more negative, and $\text{Im}(\epsilon)$ decreases.

Interestingly, in the visible wavelength range we observe a non-monotonic behavior of $\text{Im}(\epsilon)$ as a function of wavelength (Fig. 2b) that is consistent with previously reported functional dependences of $\text{Im}(\epsilon)$ on wavelength [11, 13, 14]. On the other hand, as we observe in our work, $\text{Im}(\epsilon)$ typically monotonically grows as a function of wavelength in the near-infrared wavelength range [11, 13, 14].

We added the following discussion to the SI of the manuscript:

“When analyzing ellipsometry data, we used two or three Lorentz oscillators to fit the data obtained from the fabricated films. The key issue here is that optical properties of TiN films strongly depend on the film stoichiometry (N/Ti ratio), impurities (residual oxygen, post growth oxidation), grain size (which affects the mean free path of the conduction electrons) and density/porosity (which affect the conduction electron density) [13]. All these factors likely differ from one film to another resulting in different values of fitting parameters of a Drude-Lorentz model, such as electron mobility and frequencies of Lorentz oscillators. In particular, it has been previously shown that the spectral position of the zero crossing (ENZ point) is an indicator of film stoichiometry. It has been shown that for stoichiometric TiN films this crossing occurs at

2.65 eV (468 nm) [16]. As one can see from Fig. 2a, none of our films has an ENZ crossing at 468 nm. Hence, all films reported in this work are non-stoichiometric. Moreover, it has been recently shown that amount of residual oxygen in the sputtering chamber can dramatically affect optical properties of TiN films [11, 12, 14]. Based on these reports, we expect that each fabricated film has different chemical and structural composition. The review article [13] discusses the relation between the band structure of TiN obtained via density functional theory (DFT) calculations and the spectral positions of Lorentz oscillators in an ellipsometry fit. As one can see from Table 2 of the mentioned review article [13], the spectral positions of Lorentz oscillators show significant variation. Moreover, the reported spectral position values do not always agree with DFT calculations. The reason for this is that the DFT calculations are performed for stoichiometric TiN while the films shown in our work as well as a number of films shown in the review article [13] are non-stoichiometric. Moreover, we performed compositional analyses on 135 nm-thick TiN films deposited on Si substrates via DC sputtering. The sputtering conditions were chosen to be identical to those used to deposit TiN in the tunable ENZ-TiN/SiO₂/Ag heterostructure. We found that the stoichiometry of our film is given as TiN_{0.8}O_{0.2}, where oxygen has been introduced into our film unintentionally. This result is consistent with previous reports [11, 12]. Hence, most likely, 7 nm-thick TiN films incorporated in our TiN/SiO₂/Ag films also include a significant amount of oxygen impurities. Based on this, it is not straightforward to relate positions of Lorentz oscillators to the band structure of our TiN, due to limited knowledge of the band structure of the actual film. To summarize, the choice of the model that we use to fit the ellipsometry data (two vs. three Lorentz oscillators, etc.) is largely dictated by the composition and structure of the films, which varies. Due to a limited knowledge of the dependence of the TiN band structure on the material stoichiometry and the amount of incorporated oxygen impurities, we are not able to perform a direct comparison between the positions of Lorentz oscillators and the material band structure.

One can obtain indirect evidence of compositional and structural differences in sputtered TiN films when analyzing the dependence of the imaginary part of the measured dielectric permittivity $\text{Im}(\epsilon)$ on the carrier concentration of the films. Indeed, $\text{Im}(\epsilon)$ monotonically increases with the carrier concentration for carrier concentrations less than or equal to $1.8 \times 10^{22} \text{ cm}^{-3}$ and abruptly decreases for the sample with carrier concentration $1.8 \times 10^{22} \text{ cm}^{-3}$. This kind of behavior has been previously reported in literature [11]. In the mentioned paper, $\text{Im}(\epsilon)$ of “intermediately doped” TiN is greater than $\text{Im}(\epsilon)$ of “dielectric” TiN [11]. On the other hand, over a broad wavelength range, $\text{Im}(\epsilon)$ of “metallic” TiN is smaller than $\text{Im}(\epsilon)$ of “intermediately doped” or “dielectric” TiN [11]. It is argued that when the oxygen impurity concentration in TiN is lessened, the crystallinity of TiN films is improved, $\text{Re}(\epsilon)$ becomes more negative, and $\text{Im}(\epsilon)$ decreases.

Interestingly, in the visible wavelength range we observe a non-monotonic behavior of $\text{Im}(\epsilon)$ as a function of wavelength (Fig. 2b) that is consistent with previously reported functional dependences of $\text{Im}(\epsilon)$ on wavelength [11, 13, 14]. On the other hand, as we observe in our work, $\text{Im}(\epsilon)$ typically grows monotonically as a function of wavelength in the near-infrared wavelength range [11, 13, 14].”

R3-4 The method used to record the time resolved data involves integrating over the 500-650nm spectral range. This approach is reasonable, but assumes that the form of the emission spectrum is independent of the excitation level. Did the authors check that the form of the emission spectrum was not time dependent?

We believe the reviewer’s question is related to possible distortion of PL intensity spectra occurring at the timescales smaller than the QD lifetime. Unfortunately, our setup does not allow us to record the time-dependent PL intensity profile from a single excitation-emission cycle because of the limited time resolution of our detector (50 ms) and short QD lifetimes (344–460 ps). In principle, this information can be acquired by using a streak camera which is currently not available to us.

In our experiment, we use time-correlated single photon counting (TCSPC) to measure QD lifetime [20]. Instead of collecting information from a single excitation-emission cycle, we use periodic laser excitation to extend the data collection over multiple cycles of excitation and emission. This approach allows one to construct a histogram of photon arrivals. The constructed histogram gives the number of photon counts at different photon arrival times. The time resolution of our lifetime measurement setup is 200 ps.

To clarify this point we added the following paragraph to the Methods section of the manuscript:

“We measured QD lifetime by using a time-correlated single photon counting (TCSPC) module (PicoHarp 300, PicoQuant) and single photon avalanche diode (SPAD) detector (PDM 50T, MicroPhoton Devices) [20]. We used periodic pulsed laser excitation and collect photons from multiple excitation and emission cycles. This approach allowed us to construct a histogram of number of photon counts at different photon arrival times. The time resolution of our lifetime measurement setup was 200 ps. PL lifetime decays were acquired from a particular region less than 5 μm in diameter with an iris. During lifetime measurements a 600–650 nm band-pass filter was used.”

Otherwise, if reviewer’s question is related to the reproducibility of our experiment, we would like to mention that we measured emission spectra from the sample at different times. Even though we measured

slight difference between the emission spectra depending on the part of the sample that was illuminated, we obtained reproducible PL intensity spectra when the position of the illumination spot was fixed.

To clarify this point we added the following sentence to the Methods section of the manuscript:

“Even though we measured slight differences in the emission spectrum depending on the part of the sample that was illuminated, we obtained reproducible data when the position of the illumination spot was fixed. We also verified that the shape of PL intensity spectrum does not depend on pump field intensity (Fig. S16).”

Figure S16 | Normalized PL spectra for different excitation powers. PL spectra of InP QDs embedded in a TiN/SiO₂/Ag plasmonic heterostructure. To facilitate the comparison between different PL spectra, we normalize the recorded PL spectra so that they have identical amplitude.

References

1. Dimitrov, S. & Haas, H. *Principles of LED light communications: Towards networked Li-Fi*. (2015).
2. Graydon, O. View from... ECOC 2015: Lighting that talks. *Nat. Photon.* **9**, 785–786 (2015).
3. Robertson, J., High dielectric constant oxides. *Eur. Phys. J. Appl. Phys.* **28**, 265–291 (2004).
4. Hoang, T. B., Akselrod, G. M. & Mikkelsen, M. H. Ultrafast room-temperature single photon emission from quantum dots coupled to plasmonic nanocavities. *Nano Lett.* **16**, 270–275 (2016).
5. Akselrod, G. M. *et al.* Probing the mechanisms of large Purcell enhancement in plasmonic nanoantennas. *Nat. Photon.* **8**, 835–840 (2014).
6. Wood, V. & Bulovic, V. Colloidal quantum dot light-emitting devices. *Nano Rev.* **1**, 5202 (2010).
7. Novotny, L. & Hecht, B. *Principles of nano-optics*. 2nd edn. (Cambridge University Press, 2012).
8. Tielrooij, K. J. *et al.* Electrical control of optical emitter relaxation pathways enabled by graphene. *Nat. Phys.* **11**, 281–287 (2015).
9. Reserbat-Plantey, A. *et al.* Electromechanical control of nitrogen-vacancy defect emission using graphene NEMS. *Nat. Commun.* **7**, 10218 (2016).
10. Hoang, T. B. *et al.* Ultrafast spontaneous emission source using plasmonic nanoantennas. *Nat. Commun.* **6**, 7788 (2015).
11. Zgrabik, C. M. & Hu, E. L. Optimization of sputtered titanium nitride as a tunable metal for plasmonic applications. *Opt. Mater. Express* **5**, 2786–2797 (2015).
12. Wang, Y., Capretti, A. & Dal Negro, L. Wide tuning of the optical and structural properties of alternative plasmonic materials. *Opt. Mater. Express* **5**, 2415–2430 (2015).
13. Patsalas, P., Kalfagiannis, N. & Kassavetis, S. Optical properties and plasmonic performance of titanium nitride. *Materials* **8**, 3128–3154 (2015).
14. Braic, L. *et al.* Titanium oxynitride thin films with tunable double epsilon-near-zero behaviour for nanophotonic applications. *ACS Appl. Mater. Inter.* doi:10.1021/acsami.7b07660 (2017).
15. Huang, Y. W. *et al.* Gate-tunable conducting oxide metasurfaces. *Nano Lett.* **16**, 5319–5325 (2016).
16. Logothetidis, S., Alexandrou, I. & Papadopoulos, A. In situ spectroscopic ellipsometry to monitor the process of TiN_x thin films deposited by reactive sputtering. *J. of Appl. Phys.* **77**, 1043–1047 (1995).
17. Johnson, P. B. & Christy, R. W. Optical constants of the noble metals. *Phys. Rev. B* **6**, 4370–4379 (1972).

18. Rowland, C. E. *et al.* Electric field modulation of semiconductor quantum dot photoluminescence: Insights into the design of robust voltage-sensitive cellular imaging probes. *Nano Lett.* **15**, 6848–6854 (2015).
19. Jun, Y. C., Huang, K. C. Y. & Brongersma, M. L. Plasmonic beaming and active control over fluorescent emission. *Nat. Commun.* **2**, 283 (2011).
20. Sun, S. L. *et al.* High-efficiency broadband anomalous reflection by gradient meta-surfaces. *Nano Lett.* **12**, 6223–6229 (2012).

REVIEWERS' COMMENTS:

Reviewer #1 (Remarks to the Author):

I would like to thank the authors for their very detailed and honest reply. I believe they have addressed all of the technical issues I raised.

To be honest, I am still not convinced about the future impact of such a work, given the low modulation amplitude of the emission. However, I believe it is a fair proof-of-principle of the claimed phenomenon and should be regarded as such. The idea and results could contribute to drive progresses in the field.

Therefore, I recommend this manuscript for publication in Nature Communications.

As a minor comment, there are a few typos in the paper (e.g. LODS instead of LDOS, etc..) that the authors should correct.

Reviewer #2 (Remarks to the Author):

The authors have fully addressed my concerns in the response letter and implemented them in the revised manuscript. By addressing the comments from all reviewers, the manuscript has been greatly improved. Thus, I recommend it to be published in Nature Communications.

Reviewer #3 (Remarks to the Author):

The authors have comprehensively addressed the concerns and technical questions raised in my report. I am now satisfied and can recommend that the manuscript is accepted for publication in Nature Communications